# MARS: A MALIGNITY-AWARE BACKDOOR DEFENSE IN FEDERATED LEARNING

## ABSTRACT

Federated Learning (FL) is a distributed paradigm aimed at protecting participant data privacy by exchanging model parameters to achieve high-quality model training. However, this distributed nature also makes FL highly vulnerable to backdoor attacks. Notably, the recently proposed state-of-the-art (SOTA) attack, 3DFed (SP2023), uses an indicator mechanism to determine whether the backdoor models have been accepted by the defender and adaptively optimizes backdoor models, rendering existing defenses ineffective. In this paper, we first reveal that the failure of existing defenses lies in the employment of empirical statistical measures that are loosely coupled with backdoor attacks. Motivated by this, we propose a **M**alignity-**A**ware backdoo**R** defen**S**e (MARS) that leverages backdoor energy (BE) to indicate the malicious extent of each neuron. To amplify malignity, we further extract the most prominent BE values from each model to form a concentrated backdoor energy (CBE). Finally, a novel Wasserstein distance-based clustering method is introduced to effectively identify backdoor models. Extensive experiments demonstrate that MARS can defend against SOTA backdoor attacks and significantly outperforms existing defenses.

## 1 INTRODUCTION

Federated Learning (FL) McMahan et al. (2017) is a distributed paradigm that leverages data distributed across multiple clients to train a high-quality global model without requiring data to be shared with a third party. Due to its exceptional privacy-preserving features and efficient utilization of decentralized data, FL has found widespread applications in fields such as healthcare Antunes et al. (2022), education Fachola et al. (2023), finance Chatterjee et al. (2024), and even the military Arora et al. (2024). However, the distributed nature also makes it highly susceptible to poisoning attacks Shi et al. (2022). Among these, Byzantine attacks aim to degrade the global model accuracy, while backdoor attacks trigger malicious behavior (*e.g.*, classify any input as the attacker's desired target class) only under specific conditions (*e.g.*, a white patch in the bottom right corner of an image). Because backdoor attacks do not affect the model's performance on clean samples, it is difficult for model users to realize that a backdoor has been implanted. This makes backdoor attacks a greater potential threat to FL.

To defend against backdoor attacks, the FL community has made significant efforts. Certain defenses constrain the norm of local updates to prevent backdoor updates from dominating the global model Wan et al. (2023); Wang et al. (2020); Cao et al. (2021); Wan et al. (2022). Other strategies employ out-of-distribution (OOD) detection techniques to eliminate local updates that significantly deviate from the overall distribution Nguyen et al. (2022); Zhang et al. (2022); Wang et al. (2022); Blanchard et al. (2017). Additionally, some defenses focus on detecting model consistency, such as the cosine similarity of updates, and assign lower aggregation weights to updates with high consistency (indicative of Sybil attacks) or remove them altogether Fung et al. (2020); Rieger et al. (2022). However, these defenses offer limited protection. Recently proposed state-of-the-art (SOTA) attacks can easily bypass these measures. For instance, 3DFed Li et al. (2023) uses an indicator mechanism to determine if backdoor updates are being aggregated, allowing for adaptive optimization of local models. DarkFed Li et al. (2024) and CerP Lyu et al. (2023) introduce several constraint terms that make backdoor updates resemble benign updates, exhibiting properties such as moderate magnitude, reasonable distribution, and limited consistency, making it difficult to distinguish between benign

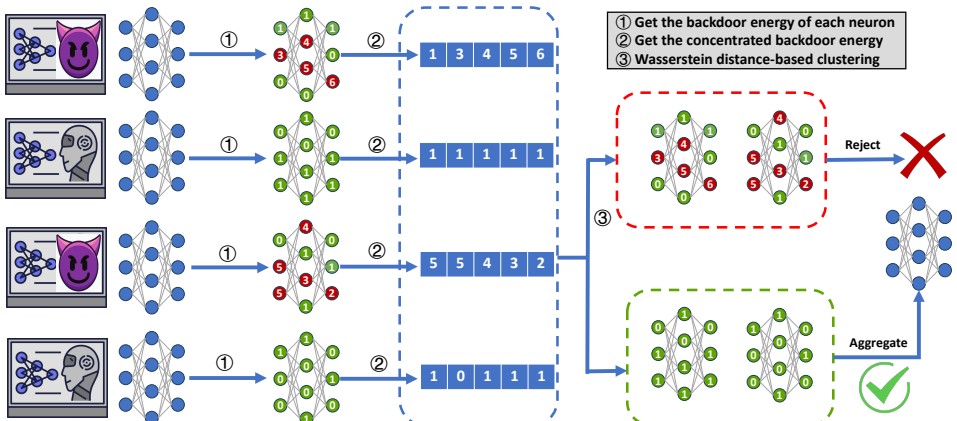

Figure 1: Overview of MARS. To facilitate understanding, we simplify the FL system to include only four clients. The first and third clients upload backdoor models, while the second and fourth clients upload benign models. Red circles represent higher backdoor energy, while green circles indicate lower backdoor energy.

and backdoor updates. These sophisticated attacks pose a significant threat to the security of FL, underscoring the urgent need for effective defenses.

In this paper, we first reveal through experimental observations that the primary statistical measures relied upon by existing defenses fail to distinguish between benign and backdoor updates when faced with SOTA attacks. We attribute this failure to the fundamental reason that ***these statistical measures are empirical and loosely coupled with backdoor attacks.*** In other words, these statistical metrics do not inherently reflect whether a local update has been compromised with a backdoor. The lack of perceiving malicious intent in existing defenses provides attackers with the opportunity to mimic the statistical distribution of benign updates, thereby defeating these defenses. Motivated by this, we propose MARS, a **M**alignity-**A**ware backdoo**R** defen**S**e. Specifically, we introduce the concept of backdoor energy (BE), which indicates the malignancy level of each neuron in the model (*i.e.*, its relevance to backdoor intent), thereby achieving a strong coupling with backdoor attacks. To amplify the malignity, we further extract the most prominent BE values in each local model to form the concentrated backdoor energy (CBE), concentrating the backdoor information. Finally, a novel Wasserstein distance-based clustering algorithm is proposed to detect backdoor models. This new clustering focuses on the probability density of elements in CBEs, thus avoiding the issues of element order sensitivity encountered by existing Euclidean and cosine distance-based clustering methods. An overview of MARS is illustrated in Figure 1.

In summary, the contributions of this paper are as follows:

- We identify the failure of existing FL backdoor defenses, attributing their failures to a reliance on empirical statistical measures that are loosely coupled with backdoor attacks. From a new perspective, we propose a robust FL defense strategy with malignity-aware capabilities.
- We introduce MARS, which detects potentially harmful neurons by incorporating the concept of backdoor energy, and we also propose a Wasserstein distance-based clustering algorithm to enhance the precise identification of backdoor models.
- We conduct extensive experiments to evaluate the effectiveness of MARS. The results demonstrate that MARS can counter SOTA backdoor attacks and consistently provide superior protection for FL compared to existing defenses.

## 2 RELATED WORK

### 2.1 BACKDOOR ATTACKS IN FEDERATED LEARNING

Since its inception, FL has been a focal point for research on backdoor vulnerabilities. Model Replacement Attack (MRA) Bagdasaryan et al. (2020), the pioneering backdoor attack on FL, works by proportionally amplifying backdoor updates to ensure that even a few malicious updates can

dominate the global model. Wang et al. (2020) later introduced the edge-case backdoor attack, leveraging rare samples from the dataset's tail to activate backdoors. Xie et al. (2020) proposed DBA, which divides a complete trigger into multiple sub-triggers assigned to different attackers to create backdoor samples, aiming to reduce the pairwise similarity of malicious updates. However, these early attacks often neglected the possibility of defensive measures, rendering them effective primarily against FL systems with no defenses or only weak ones.

To address this shortcoming, a new wave of sophisticated backdoor attacks has emerged. 3DFed Li et al. (2023), for instance, employs an indicator mechanism to detect whether backdoor updates are being aggregated and then adaptively optimizes the backdoor models to evade existing defenses. Similarly, CerP Lyu et al. (2023) and DarkFed Li et al. (2024) share a core strategy of adding constraints to mimic the characteristics of benign updates—such as moderate magnitudes and limited consistency—striking a balance between stealth and efficacy. These advanced attacks significantly threaten the secure deployment of FL, necessitating the development of robust defenses.

## 2.2 BACKDOOR DEFENSES IN FL

We broadly categorize existing defenses into three main types based on the techniques they employ: norm constraint-based defenses, OOD detection-based defenses, and consistency detection-based defenses.

Norm constraint-based defenses posit that the optimal point for the backdoor task typically deviates significantly from the optimal point for the main task. This results in the norm of backdoor updates being much larger than that of benign updates. Consequently, these defenses constrain the norm of all local updates within a reasonable range. Norm Clipping Wang et al. (2020) serves as a representative example of such defenses. Additionally, some other defenses Wan et al. (2023; 2022); Cao et al. (2021) also leverage this characteristic to prevent malicious updates from dominating the global model.

OOD detection-based defenses assert that backdoor updates and benign updates exhibit substantial differences in their distributions, with benign updates typically being densely distributed. In contrast, backdoor updates can be considered as outliers. Building on this premise, Multi-Krum Blanchard et al. (2017) calculates an anomaly score for each local update based on the sum of its distances to its neighboring nodes. A higher score indicates greater deviation, making it more likely to be discarded. RFLBAT Wang et al. (2022) utilizes Principal Component Analysis (PCA) to project local updates into a low-dimensional space. Subsequently, it employs a clustering algorithm to identify outliers, marking them as backdoor updates. FLAME Nguyen et al. (2022) identifies updates that deviate significantly in direction from the overall trend as backdoor updates and excludes them from the aggregation queue. FLDetector Zhang et al. (2022) exploits the differences between the predicted model and the actual model to discover outliers.

Consistency detection-based defenses argue that all backdoor updates share the same objective, namely, to classify trigger-carrying samples as the target label. Therefore, these updates exhibit strong consistency, either in terms of update directions or neuron activations. On the other hand, diverse benign updates may display lower consistency due to data heterogeneity Li et al. (2020). With this understanding, FoolsGold Fung et al. (2020) assigns lower aggregation weights to updates with high pairwise cosine similarities, thereby mitigating the impact of backdoor updates. DeepSight Rieger et al. (2022) uses the consistency on neuron activations in the backdoor model to detect malicious updates.

## 3 THREAT MODEL

### 3.1 ATTACK MODEL

The primary objective of the attackers is to implant a backdoor into the global model by transmitting malicious model parameters to the central server. To facilitate more sophisticated backdoor attacks, we assume the attackers possess substantial capabilities:

- **Flexible Local Optimization.** Attackers can arbitrarily modify their local optimization objectives, achieving a fine balance between stealth and effectiveness.

- **Collusion Capability.** Attackers can collude, allowing full transparency of training data and model parameters among them. This transparency aids in dynamically adjusting backdoor models to evade defense mechanisms.

- **Dominant Presence.** Attackers can constitute a majority, with their proportion not restricted to below $50\%$ as typically assumed in existing research.

These powerful assumptions significantly heighten the challenge of defending against backdoor attacks.

## 3.2 DEFENSE MODEL

Our proposed defense is deployed at the central server to detect and filter out backdoor models from the local models uploaded by clients, resulting in a high-performance, backdoor-free global model. We assume the central server has minimal knowledge. Specifically, the server only has access to the model parameters of all local models in each round. It cannot access any client's training data or control the training process of any client's model. Moreover, the server does not make any assumptions about the proportion of attackers. Our proposed defense algorithm aims to achieve the following goals simultaneously:

- **Effectiveness.** Regardless of the type of backdoor attack, the defense should effectively thwart the attackers' malicious activities, resulting in a backdoor-free global model.

- **Practicability.** The defense should remain effective in challenging real-world scenarios, such as when the proportion of attackers exceeds $50\%$, clients have heterogeneous data distributions, or clean auxiliary datasets are unavailable.

- **Fidelity.** In non-adversarial scenarios (*i.e.*, there are no attackers in the FL system), the accuracy of the global model on clean samples should not degrade compared to FedAvg due to the deployment of this defense.

These objectives ensure that the defense is robust, practical, and reliable in both adversarial and non-adversarial environments.

# 4 MARS

## 4.1 MOTIVATION

After reviewing the SOTA defenses, we find that they mainly rely on empirical statistical measures. Techniques such as norm constraint, OOD detection, and consistency detection are extensively utilized by them. For example, methods like Norm Clipping Wang et al. (2020), FLAME Nguyen et al. (2022), and DeepSight Rieger et al. (2022) constrain local updates within a hyperball of normal radius to prevent backdoor updates with excessive magnitudes from dominating the global model. Defenses like Mulri-Krum Blanchard et al. (2017), RFLBAT Wang et al. (2022), FLDetector Zhang et al. (2022), and FLAME Nguyen et al. (2022) exclude local updates that significantly deviate from the overall distribution or the predicted global update. Moreover, DeepSight Rieger et al. (2022) and FoolsGold Fung et al. (2020) assume that backdoor updates share a common objective—classifying samples with triggers into a predetermined target class—resulting in higher consistency, particularly in terms of cosine similarity. Based on this, these defenses assign lower aggregation weights to highly consistent updates or exclude them entirely. It's worth noting that some composite defenses, such as FLAME and DeepSight, span multiple statistical measures so as to enhance the robustness against backdoor attacks. However, we demonstrate that these empirical statistical measures tend to fail when faced with advanced attacks.

**Failure of Norm Constraint.** To prevent the norm (also known as magnitude) of backdoor updates from becoming excessively large, some advanced backdoor attacks Li et al. (2024; 2023); Lyu et al. (2023) incorporate a constraint term to encourage finding backdoor models near the previous round's global model. The resulting backdoor updates, even without proportionally increasing their magnitude, can still achieve excellent attack efficacy. As shown in Figure 2(a), the magnitude of backdoor updates obtained this way can be even smaller than that of benign updates. This indicates that when

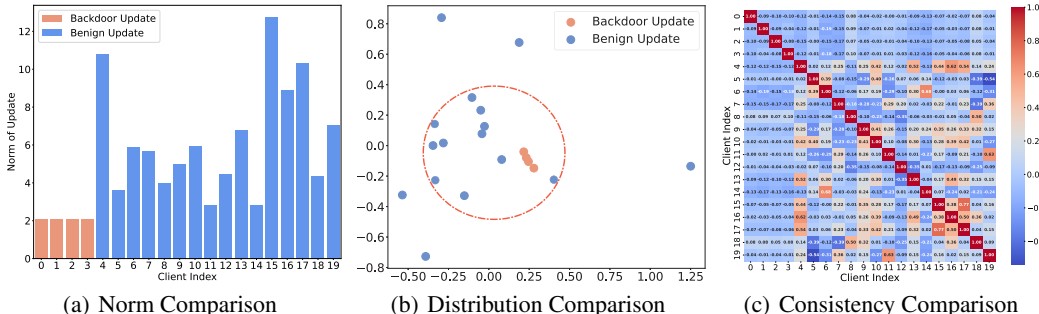

Figure 2: Comparison of statistical measures between backdoor and benign updates. We consider 20 clients, with the first 4 clients (indices 0 to 3) being malicious and conducting 3DFed attack, while the remaining clients (indices 4 to 19) are benign. (a) provides the norms of all local updates. (b) shows the distribution of all local updates projected into 2D space using PCA. (c) presents a heatmap of the similarities between local updates.

a defender employs the norm constraint, backdoor updates remain unaffected. Consequently, this type of statistical measure can be easily bypassed by these advanced backdoor attacks.

**Failure of OOD Detection.** To make backdoor updates appear less anomalous, several advanced backdoor attacks have devised innovative solutions. 3DFed Li et al. (2023) generates a series of outlier decoy updates, making the backdoor updates seem more benign in comparison, thus bypassing the detection by a defender. DarkFed Li et al. (2024) adds a constraint term to ensure that the cosine similarity between backdoor and benign updates is close to that among benign updates themselves. CerP Lyu et al. (2023) employs a similar strategy to DarkFed but uses Euclidean distance for constraint. As shown in Figure 2(b), the backdoor updates crafted by 3DFed are indistinguishable from benign ones when projected onto a 2-dimensional space. Moreover, Figure 2(c) provides new evidence from another perspective. It illustrates that the cosine similarity between backdoor and benign updates hovers around $-0.08$, which is even higher than the similarity among some benign updates (*e.g.*, a cosine similarity of $-0.54$ between client 5 and client 19). Consequently, OOD detection fails to provide effective protection against these attacks.

**Failure of Consistency Detection.** To reduce the consistency of backdoor updates, 3DFed Li et al. (2023) adds carefully designed noise masks to each backdoor update, increasing the variability among them without diminishing the strength of the attack. DarkFed Li et al. (2024) and CerP Lyu et al. (2023) achieve a similar effect by adding a constraint term to decrease the cosine similarity between pairs of backdoor updates. As shown in Figure 2(c), the cosine similarity between backdoor updates is only about $-0.08$, which is significantly lower than the cosine similarity among some benign updates (*e.g.*, $0.77$ between client 15 and client 17). This indicates that consistency detection also fails to differentiate between backdoor and benign updates.

We attribute all the aforementioned failures to a fundamental reason: ***these empirical statistical measures are loosely coupled with backdoor attacks.*** In other words, they lack the capability to perceive malicious intent and do not fundamentally reflect whether an update has been compromised with a backdoor. Consequently, attackers can easily mimic the statistical measures of benign updates, thereby bypassing existing defenses. This motivates us to employ a malignity-aware measure that can reflect the inherent maliciousness of the model, rather than relying on empirical intuitions.

## 4.2 OVERVIEW OF MARS

Unlike existing schemes that directly detect abnormal statistical measures based on model parameters, we propose a **M**alignity-**A**ware backdoo**R** defen**S**e (MARS). For each local model, we first calculate the *backdoor energy* (BE) of each neuron, which reflects how strongly a neuron is associated with backdoor attacks. Higher backdoor energy indicates a higher level of malignity for that neuron. To further amplify the malignity, we extract the most prominent backdoor energies from each layer and concatenate them into a one-dimensional vector, which we call the *concentrated backdoor energy* (CBE). Note that CBE is not unique to backdoor models; it can also be calculated for benign models. Subsequently, we propose a novel Wasserstein-based clustering method to effectively identify backdoor models and prevent them from participating in the aggregation.

## 4.3 Obtaining Backdoor Energy

Given a $L$-layer neural network $F^1 = f^{(L)} \circ f^{(L-1)} \circ \ldots \circ f^{(1)}$, a clean dataset $D \subseteq \mathcal{X} \times \mathcal{Y}$, and a backdoor trigger generator $\delta(.)$, a straightforward way to evaluate the backdoor energy of the $k^{th}$ neuron in the $l^{th}$ layer is to compute the expected difference in neuron values between clean samples and backdoor samples:

$$BE_k^{(l)}(F) = \mathbb{E}_{x \sim \mathcal{X}} \left\| F_k^{(l)}(x) - F_k^{(l)}(\delta(x)) \right\|_2, \tag{1}$$

where $F_k^{(l)}(.) = f_k^{(l)} \circ f^{(l-1)} \circ \ldots \circ f^{(1)}(.)$ indicates the neuron function that maps an input sample to the $k^{th}$ neuron in the $l^{th}$ layer.

However, obtaining BE via equation 1 faces two challenges. First, due to the privacy-preserving nature of FL, the clean dataset $D$ is inaccessible to the defender. Furthermore, the trigger is very subtle and private to the attackers, making it difficult for the defender to obtain. A naive idea is to collect a shadow dataset and employ reverse engineering Wang et al. (2019) to reconstruct the trigger. However, the shadow dataset may significantly different from the real training dataset, leading to inaccurate BE calculations and impairing the detection of backdoor models. Moreover, reverse engineering requires reconstructing a trigger for each class individually, which becomes very time-consuming when there are many classes (*e.g.*, ImageNet with 1000 classes). Additionally, when the trigger is complex, the reconstructed trigger may significantly differ from the real one, also resulting in inaccurate BE calculations. Considering the above challenges, we turn to exploring the upper bound of BE.

**Theorem 1** (**Upper Bound of Backdoor Energy**). *Suppose a L-layer neural network $F$ and its every sub-network $f^{(l)}, l \in [1, L]$, are Lipschitz smooth. Then, the backdoor energy of the $k^{th}$ neuron in the $l^{th}$ layer can be upper bounded by:*

$$BE_k^{(l)}(F) \leq \|f_k^{(l)}\|_{Lip} \prod_{i=1}^{l-1} \|f^{(i)}\|_{Lip} \mathbb{E}_{x \sim \mathcal{X}} \|x - \delta(x)\|_2, \tag{2}$$

*where $\|.\|_{Lip}$ represents the Lipschitz constant of a function. The detailed proof is provided in Appendix A.1.*

On the one hand, we do not need the exact value of BE for subsequent calculations, but only the relative magnitudes of BE among different neurons to detect anomalies. On the other hand, the upper bound of backdoor energy reasonably reflects the distribution of BE. Thus we can approximate $BE$ using its upper bound. Furthermore, as indicated by formula 2, when considering different neurons $j$ and $k$ in the same layer $l$, the difference in their $BE$ upper bounds is solely in the first term, *i.e.*, $\|f_j^{(l)}\|_{Lip}$ and $\|f_k^{(l)}\|_{Lip}$. Therefore, we can further approximate BE using only the first term of its upper bound:

$$BE_k^{(l)}(F) = \|f_k^{(l)}\|_{Lip}. \tag{3}$$

Notably, equation 3 does not rely on the clean dataset or the trigger. It allows for the easy calculation of BE for all neurons using only the model parameters.

## 4.4 Obtaining Concentrated Backdoor Energy

Since a backdoor can be viewed as a shortcut Wang et al. (2019), only a small number of neurons are backdoor-related. Therefore, we extract the highest BE values from each layer (*e.g.*, the top $5\%$ by default in our paper) and concatenate them into a one-dimensional vector. We call this vector the concentrated backdoor energy (CBE), as it aggregates the most prominent backdoor energies in the model. This approach minimizes interference from neurons unrelated to the backdoor, aiding in the subsequent differentiation between backdoor and benign models. Formally, the CBE of a model $F$ can be obtained by:

$$CBE(F) = \bigcup_{l=1}^{L} TopK_{\kappa\%} \left( \{BE_i^{(l)}(F))\}_{i=1}^{n_l} \right), \tag{4}$$

where $L$ is the total number of layers, $n_l$ is the number of neurons in the $l^{th}$ layer, $TopK_{\kappa\%}(.)$ denotes the top $\kappa\%$ values of a set.

---

[1]We omit the model weights $\theta$ in $F(.; \theta)$ for simplicity.

### 4.5 IDENTIFYING BACKDOOR MODELS

CBE can effectively capture each local model's backdoor information, in which backdoor and benign models are quite different, making clustering a promising approach for identifying backdoor models. However, two challenges remain to be addressed. First, existing clustering methods primarily use Euclidean distance or cosine distance as metrics, which are highly sensitive to the order of elements rather than their overall distribution, leading to potential errors. Second, after clustering, it is challenging to decide which clusters to trust and include in the final aggregation. Choosing the wrong clusters could result in the failure to exclude backdoor models. Compulsively discarding some clusters, in innocent scenario (*i.e.*, all clients are benign), may slow down the convergence of the global model or even decrease its accuracy.

**Wasserstein Distance-Based Clustering**. We use K-Means to partition the CBEs of all local models into two clusters. However, the default metric, Euclidean distance, or the widely used cosine distance[2], is sensitive to the order of elements and not suitable for our scenario. This is particularly true for FL, where the top BE values of different local backdoor models may appear in different neurons

Table 1: Metric comparison.

| **Metric** | (**L1**, **L2**) | (**L1**, **L3**) | (**L2**, **L3**) |
|---|---|---|---|
| **Euc.** | 6.16 | **5.48** | 6.16 |
| **Cos.** | 0.31 | 0.10 | **0.07** |
| **Wass.** | **0.40** | 2.00 | 2.40 |

due to the distributed nature of training. As a result, even though the elements in the CBEs of backdoor models are generally larger, both Euclidean and cosine distances do not recognize these CBEs as similar. To focus on the distribution of elements in the CBE and avoid the influence of their order, we employ Wasserstein distance Panaretos & Zemel (2019) as the metric for K-Means and call the clustering algorithm K-WMeans. Formally, for two probability distributions $p$ and $q$, the Wasserstein distance between them is defined as:

$$Wass(p, q) = \inf_{\gamma \sim \prod(p,q)} \mathbb{E}_{x,y \sim \gamma} \|x - y\|, \tag{5}$$

where $\prod(p, q)$ denotes the set of all possible joint distributions between $p$ and $q$. Next, we use a toy example to demonstrate that Wasserstein distance is more suitable than Euclidean and cosine distances for identifying backdoor models in our case.

***Toy Example:*** *Assume $L1 = [1, 3, 4, 5, 6]$ and $L2 = [5, 5, 4, 3, 2]$ are the CBEs of backdoor models, and $L3 = [1, 1, 1, 1, 1]$ is the CBE of a benign model. This assumption is reasonable because neurons in backdoor models have higher BE values. As shown in Table 1, when considering Euclidean distance, $L1$ and $L3$ are deemed the closest. When using cosine distance as the metric, $L2$ and $L3$ are considered the closest. Both metrics are not conducive to clustering backdoor models into a single cluster. Notably, when considering Wasserstein distance, despite the significant differences in values across each dimension for $L1$ and $L2$, their distance is much smaller than the distances between $L1$ and $L3$ or $L2$ and $L3$. This favors the clustering of backdoor models together.*

**Cluster Selection.** After using K-WMeans to divide the CBEs into two clusters, the subsequent challenge is how to select the trusted cluster. Existing methods typically assume that benign clients are the majority and therefore accept the larger cluster. However, in some extreme scenarios, the number of attackers might exceed that of benign clients, leading to the unintended selection of backdoor models. To avoid this assumption, we use the norm of the cluster center as a more reliable metric for cluster selection. Specifically, the elements in the CBEs of attackers generally have higher values than those in the CBEs of benign clients. Therefore, we select the cluster with the smaller center norm, rather than relying on the majority.

However, when there are no attackers in the FL system, blindly discarding the cluster with the larger center norm might slow down the global model's convergence or even reduce its accuracy. To address this, we use the Wasserstein distance to measure the similarity between clusters. If the distance between the two clusters does not exceed a threshold $\epsilon$, it indicates that the CBEs of all local models have similar distributions, corresponding to a scenario where all local models are either benign or malicious. Given that an FL system with only attackers is meaningless, we assume that when the cluster distance is low, all local models are benign. Therefore, in this case, both clusters are selected. A detailed algorithm description is provided in Appendix A.2.

---

[2]One minus cosine similarity.

## 5 EXPERIMENTS

### 5.1 EXPERIMENTAL SETUP

We consider an FL system with 100 clients, where 20 of them are designated as attackers. In each round, 20 clients are selected to participate in the FL process, with 4 of them guaranteed to be attackers. By default, MARS's hyperparameters $\kappa$ and $\epsilon$ are set to 5 and 0.03, respectively.

**Datasets, models, and codes.** We evaluate the effectiveness of MARS on MNIST LeCun et al. (1998), CIFAR-10 Krizhevsky & Hinton (2009), and CIFAR-100 Krizhevsky & Hinton (2009) datasets. To simulate realistic non-IID distributions, we use the Dirichlet distribution with a default sampling parameter $\alpha$ set to 0.9. For MNIST, a simple CNN is employed as the global model, while for CIFAR-10 and CIFAR-100, we use ResNet-18 He et al. (2016) as the global model. The code will be made publicly available upon the publication of the paper.

**Evaluated attacks and defenses.** We consider three SOTA backdoor attacks: MRA Bagdasaryan et al. (2020), CerP Lyu et al. (2023), and 3DFed Li et al. (2023). Additionally, we design a customized adaptive attack tailored specifically for MARS. On the defense side, we evaluate eight SOTA defense methods, including FedAvg McMahan et al. (2017), Multi-Krum Blanchard et al. (2017), RFLBAT Wang et al. (2022), FLAME Nguyen et al. (2022), FoolsGold Fung et al. (2020), FLDetector Zhang et al. (2022), Deepsight Rieger et al. (2022), and FedCLP Zheng et al. (2022). Notably, we also include the recently published backdoor defense, BackdoorIndicator Li & Dai (2024), from Usenix Security 2024. Detailed descriptions of these attacks and defenses can be found in Appendix A.3.

**Evaluation metrics.** We assess the performance of the defenses using several metrics, including model accuracy (ACC), attack success rate (ASR), true positive rate (TPR), false positive rate (FPR), and comprehensive ability of defense (CAD). Higher values of ACC, TPR, and CAD, along with lower values of ASR and FPR, indicate a more effective defense. A more detailed description to these metrics can be found in Appendix A.4.

Table 2: Comparison of MARS and SOTA defenses under SOTA attacks.

| Dataset | Baselines | MRA | | | | | CerP | | | | | 3DFed | | | | |
|---|---|---|---|---|---|---|---|---|---|---|---|---|---|---|---|---|
| | | ACC ↑ | ASR ↓ | TPR ↑ | FPR ↓ | CAD ↑ | ACC ↑ | ASR ↓ | TPR ↑ | FPR ↓ | CAD ↑ | ACC ↑ | ASR ↓ | TPR ↑ | FPR ↓ | CAD ↑ |
| MNIST | FedAvg | 98.46 | 99.67 | 0.00 | 0.00 | 49.70 | 99.08 | 88.32 | 0.00 | 0.00 | 52.69 | 98.96 | 77.17 | 0.00 | 0.00 | 55.45 |
| | Multi-Krum | 98.97 | 9.73 | 100.00 | 0.00 | 97.31 | 99.34 | 9.76 | 100.00 | 0.00 | 97.40 | 99.08 | 85.06 | 0.00 | 25.00 | 47.26 |
| | RFLBAT | 98.98 | 9.71 | 100.00 | 19.38 | 92.47 | 90.84 | 21.00 | 90.00 | 16.88 | 85.74 | 99.13 | 74.53 | 0.00 | 18.75 | 51.46 |
| | FLAME | 98.98 | 9.63 | 100.00 | 31.25 | 89.53 | 99.31 | 9.74 | 100.00 | 31.25 | 89.58 | 98.65 | 91.41 | 0.00 | 56.25 | 37.75 |
| | FoolsGold | 99.00 | 9.64 | 100.00 | 0.00 | 97.34 | 99.31 | 29.21 | 30.00 | 0.00 | 75.03 | 99.02 | 73.08 | 0.00 | 0.00 | 56.49 |
| | FLDetector | 96.61 | 94.70 | 0.00 | 17.50 | 46.10 | 99.02 | 80.61 | 10.00 | 0.00 | 57.10 | 98.73 | 74.01 | 0.00 | 0.00 | 56.18 |
| | DeepSight | 98.92 | 22.83 | 0.00 | 6.25 | 67.46 | 99.29 | 9.75 | 100.00 | 37.50 | 88.01 | 98.74 | 62.30 | 0.00 | 6.25 | 57.55 |
| | FedCLP | 82.00 | 14.47 | - | - | 83.77 | 99.21 | 9.75 | - | - | 94.73 | 85.54 | 16.69 | - | - | 84.43 |
| | **MARS** | 99.01 | 9.66 | 100.00 | 0.00 | 97.34 | 99.32 | 9.74 | 100.00 | 0.00 | 97.40 | 99.13 | 9.72 | 100.00 | 3.62 | 96.45 |
| CIFAR-10 | FedAvg | 78.32 | 99.68 | 0.00 | 0.00 | 44.66 | 84.49 | 93.70 | 0.00 | 0.00 | 47.70 | 84.37 | 96.76 | 0.00 | 0.00 | 46.90 |
| | Multi-Krum | 85.21 | 9.69 | 100.00 | 0.00 | 93.88 | 85.32 | 10.01 | 100.00 | 0.00 | 93.83 | 84.07 | 97.27 | 0.00 | 25.00 | 40.45 |
| | RFLBAT | 85.13 | 9.33 | 97.50 | 1.25 | 93.01 | 85.20 | 10.39 | 100.00 | 0.00 | 93.70 | 84.30 | 92.02 | 0.00 | 5.00 | 46.82 |
| | FLAME | 84.87 | 8.74 | 100.00 | 31.25 | 86.22 | 85.34 | 10.59 | 100.00 | 31.25 | 85.88 | 83.06 | 97.50 | 2.50 | 55.63 | 33.11 |
| | FoolsGold | 85.06 | 9.71 | 100.00 | 12.50 | 90.71 | 85.00 | 91.00 | 0.00 | 0.00 | 48.50 | 84.11 | 96.29 | 0.00 | 0.25 | 46.89 |
| | FLDetector | 85.16 | 9.96 | 100.00 | 0.00 | 93.80 | 85.18 | 88.64 | 50.00 | 0.00 | 62.39 | 84.24 | 95.20 | 0.00 | 35.00 | 38.51 |
| | DeepSight | 83.99 | 99.94 | 0.00 | 6.25 | 44.45 | 85.22 | 74.15 | 10.00 | 40.00 | 45.27 | 84.80 | 98.85 | 0.00 | 6.25 | 44.93 |
| | FedCLP | 75.01 | 10.88 | - | - | 82.07 | 78.52 | 11.00 | - | - | 83.76 | 69.25 | 7.55 | - | - | 80.85 |
| | **MARS** | 85.16 | 9.40 | 100.00 | 0.00 | 93.94 | 85.37 | 10.03 | 100.00 | 0.00 | 93.84 | 85.07 | 9.86 | 100.00 | 0.00 | 93.80 |
| CIFAR-100 | FedAvg | 77.97 | 100.00 | 0.00 | 0.00 | 44.49 | 78.87 | 99.97 | 0.00 | 0.00 | 44.73 | 78.89 | 100.00 | 0.00 | 0.00 | 44.72 |
| | Multi-Krum | 79.34 | 0.97 | 100.00 | 0.00 | 94.59 | 79.67 | 1.14 | 100.00 | 0.00 | 94.63 | 78.36 | 100.00 | 0.00 | 25.00 | 38.34 |
| | RFLBAT | 79.46 | 0.97 | 100.00 | 15.00 | 90.89 | 79.50 | 1.15 | 100.00 | 0.63 | 94.43 | 78.89 | 100.00 | 0.00 | 18.75 | 40.04 |
| | FLAME | 79.63 | 0.95 | 100.00 | 31.25 | 86.86 | 79.56 | 1.20 | 100.00 | 31.25 | 86.78 | 79.27 | 1.00 | 100.00 | 31.25 | 87.76 |
| | FoolsGold | 79.54 | 0.98 | 100.00 | 0.00 | 94.64 | 79.59 | 1.16 | 100.00 | 0.00 | 94.61 | 79.01 | 100.00 | 0.00 | 0.00 | 44.75 |
| | FLDetector | 78.10 | 100.00 | 0.00 | 0.00 | 44.53 | 78.57 | 90.10 | 10.00 | 0.00 | 49.62 | 78.16 | 100.00 | 0.00 | 50.00 | 42.04 |
| | DeepSight | 78.85 | 61.30 | 0.00 | 6.25 | 52.83 | 79.18 | 1.20 | 97.50 | 56.88 | 79.65 | 78.91 | 10.59 | 20.00 | 25.00 | 65.83 |
| | FedCLP | 77.96 | 0.91 | - | - | 88.53 | 78.36 | 1.19 | - | - | 88.59 | 77.73 | 0.99 | - | - | 88.37 |
| | **MARS** | 79.53 | 0.97 | 100.00 | 0.00 | 94.64 | 79.73 | 1.15 | 100.00 | 0.00 | 94.65 | 79.37 | 0.97 | 100.00 | 0.00 | 94.60 |

## 5.2 EXPERIMENTAL RESULTS

**Comparison with SOTA defenses.** Table 2 compares the performance of MARS with 8 SOTA defenses against 3 SOTA backdoor attacks across 5 evaluation metrics on 3 datasets. Overall, existing defenses fail to provide adequate protection, especially when confronted with advanced attacks like 3DFed. In contrast, our proposed MARS consistently achieves the best performance across all datasets and attack scenarios, demonstrating its robustness in maintaining model performance in the presence of backdoor attacks. Specifically, for the MRA attack, defenses such as Multi-Krum, RFLBAT, FLAME, and FoolsGold achieve satisfactory ASR, but they suffer from excessive client exclusion. For instance, FLAME shows an FPR as high as $31.25\%$ on the CIFAR-10 dataset. For the CerP attack, the effectiveness of existing defenses varies significantly across datasets. For example, FoolsGold can precisely detect all backdoor models on CIFAR-100, but only a few on MNIST, while its ability to detect backdoors completely breaks down on CIFAR-10. For 3DFed, most defenses show consistently high ASR, with the only exception being FedCLP, which achieves a relatively lower ASR, indicating some level of backdoor mitigation. However, FedCLP's aggressive pruning of local models often leads to excessive removal, negatively impacting the model's utility. When benchmarked against MARS, FedCLP's ACC drops by $1.64\% \sim 15.82\%$ across different datasets, a decline that is unacceptable for most real-world scenarios. We attribute MARS's superior defense ability to its detection of anomalies through BE, which is strongly correlated with backdoor attacks, fundamentally distinguishing it from existing defenses that rely on loosely coupled, empirical metrics.

**Resilience to adaptive attack.** To further assess the robustness of MARS, we consider a more informed adversary, where attackers have prior knowledge that the central server employs MARS as the defense mechanism. Leveraging this insight, the attackers can craft adaptive strategies specifically designed to bypass MARS. Since MARS detects backdoor models through their relatively higher backdoor energy, a straightforward approach for executing an adaptive attack is to introduce a regularization term to minimize the backdoor energy of each neuron in each backdoor model. Formally, the attackers' optimization objective is defined as follow:

Table 3: Performance of MARS against adaptive attack.

| $\lambda$ | Defense | ACC ↑ | ASR ↓ | TPR ↑ | FPR ↓ | CAD ↑ |
|---|---|---|---|---|---|---|
| 0.0001 | **MARS** | 85.31 | 9.43 | 100.00 | 0.00 | 93.97 |
| | **MARS*** | 85.45 | 9.86 | 100.00 | 0.00 | 93.90 |
| 0.001 | **MARS** | 85.18 | 9.75 | 100.00 | 0.00 | 93.86 |
| | **MARS*** | 85.05 | 9.44 | 100.00 | 0.00 | 93.90 |
| 0.01 | **MARS** | 85.26 | 9.57 | 100.00 | 0.00 | 93.92 |
| | **MARS*** | 85.50 | 9.76 | 100.00 | 0.00 | 93.94 |
| 0.05 | **MARS** | 10.00 | 100.00 | 0.00 | 97.50 | 3.13 |
| | **MARS*** | 85.12 | 9.30 | 100.00 | 0.00 | 93.96 |
| 0.1 | **MARS** | 10.00 | 100.00 | 0.00 | 99.38 | 2.66 |
| | **MARS*** | 85.14 | 9.31 | 100.00 | 0.00 | 93.96 |

$$\min_{\theta} \mathbb{E}_{(x,y)\sim\hat{D}} \left[ \mathcal{L}_{\text{CE}}\left(F(x;\theta), y\right) \right] + \lambda \sum_{l \in L} \sum_{k \in n_l} BE_k^{(l)}(F(.;\theta)), \tag{6}$$

where $\mathcal{L}_{\text{CE}}$ denotes the cross-entropy loss function, $\hat{D}$ consists of both clean and backdoor samples, and $\lambda$ represents the regularization coefficient. As shown in Table 3, when $\lambda$ is set to 0.01 or lower, MARS can effectively defend against adaptive attacks, achieving a CAD of over $93\%$. We hypothesize that this is due to the small regularization coefficient, which provides limited constraint on the backdoor energy of neurons. However, when $\lambda$ is further increased to 0.05 or higher, the backdoor energy of malicious models becomes sufficiently constrained, even falling below that of benign models. This causes MARS to misclassify all benign models as malicious, and vice versa[3], as indicated by a TPR of $0\%$ and an FPR close to $100\%$. Nevertheless, these results also suggest that even with constrained backdoor energy, there remain significant differences between the CBE distributions of backdoor and benign models. Therefore, we modify MARS's cluster selection strategy from choosing the cluster with the smaller center norm to a majority-based selection, which we refer to as MARS*. We observe that regardless of the $\lambda$ value, MARS* consistently and effectively defends against adaptive attacks.

---

[3]Note that the ACC also drops to $10\%$ in this case, indicating that the usability of the global model is compromised. This violates the stealthiness requirement of a backdoor attack, and therefore cannot be considered a successful backdoor attack.

**Comparison with BackdoorIndicator.** The most recently proposed defense BackdoorIndicator identifies that subsequent backdoor injections significantly slow down the ASR decline of previously implanted backdoors. Building on this observation, it employs an indicator task that uses OOD samples to detect and remove backdoored models. As shown in Table 4, BackdoorIndicator effectively detects most backdoor models under the MRA attack, maintaining a low ASR. However, when confronted with the CerP attack, it can only detect a limited number of backdoor models, resulting in an ASR close to 72%, indicating that BackdoorIndicator fails to provide sufficient protection in this case. Against the 3DFed attack, sim-

Table 4: Comparison of MARS and Back-doorIndicator. G and C100 refer to the use of GT-SRB and CIFAR-100 as the indicator datasets of BackdoorIndicator, respectively.

| Attack | Defense | ACC ↑ | ASR ↓ | TPR ↑ | FPR ↓ | CAD ↑ |
|---|---|---|---|---|---|---|
| MRA | **Indicator (G)** | 85.28 | 9.32 | 97.50 | 0.00 | 93.37 |
| | **Indicator (C100)** | 85.43 | 10.29 | 90.00 | 0.00 | 91.29 |
| | **MARS** | 85.16 | 9.40 | 100.00 | 0.00 | 93.94 |
| CerP | **Indicator (G)** | 85.22 | 71.94 | 37.50 | 0.63 | 62.54 |
| | **Indicator (C100)** | 84.89 | 71.98 | 47.50 | 0.63 | 64.95 |
| | **MARS** | 85.37 | 10.03 | 100.00 | 0.00 | 93.84 |
| 3DFed | **Indicator (G)** | 83.77 | 96.65 | 0.00 | 53.75 | 33.34 |
| | **Indicator (C100)** | 84.39 | 97.93 | 0.00 | 17.50 | 42.24 |
| | **MARS** | 85.07 | 9.86 | 100.00 | 0.00 | 93.80 |

ilar to other evaluated SOTA defenses, BackdoorIndicator completely breaks down, achieving less than half the CAD of MARS. We hypothesize that this is because BackdoorIndicator is a heuristic algorithm that validates its intuition based solely on unconstrained backdoor training. As a result, it performs well against attacks like MRA, which rely solely on data poisoning, a finding supported by both the original paper and our experimental results. However, CerP and 3DFed introduce various constraints during the backdoor model training process, making the attacks more subtle and potent. These constraints likely lead to failures in BackdoorIndicator's underlying intuition, rendering it less effective against these more sophisticated attacks.

**Impact of attacker ratio on MARS.** Previously, we demonstrated that MARS consistently outperforms existing defenses in terms of resilience to various attacks with a 20% attacker proportion (*i.e.*, the effectiveness goal). To further investigate MARS's robustness, it is important to explore how it performs across a broader range of attacker proportions. Specifically, we aim to examine if MARS mistakenly excludes benign models when there are no attackers (*i.e.*, the fidelity goal) and whether it can still provide

Table 5: Impact of attacker ratio on MARS.

| Atk. Ratio | ACC ↑ | ASR ↓ | TPR ↑ | FPR ↓ | CAD ↑ |
|---|---|---|---|---|---|
| **0** | 85.26 | 9.34 | 100.00 | 0.00 | 93.98 |
| **10** | 85.21 | 9.42 | 100.00 | 0.00 | 93.95 |
| **20** | 85.07 | 9.86 | 100.00 | 0.00 | 93.80 |
| **30** | 85.13 | 9.47 | 100.00 | 0.00 | 93.92 |
| **50** | 84.95 | 9.59 | 100.00 | 0.00 | 93.84 |
| **70** | 84.83 | 10.54 | 100.00 | 0.00 | 93.57 |
| **95** | 82.99 | 11.42 | 100.00 | 0.00 | 92.89 |

effective defense when the attacker proportion exceeds 50% (*i.e.*, the practicability goal). Table 5 presents the evaluation metrics of MARS as the attacker proportion increases from 0% to 95%. Remarkably, MARS consistently identifies all attackers with a TPR of 100%, while ensuring no benign models are misclassified as malicious (FPR of 0%) across all settings. We attribute MARS's outstanding performance in extreme scenarios (*e.g.*, 0% and 95% attacker presence) to its carefully designed cluster selection strategy, which utilizes the cluster center norm to identify the trusted cluster and decides whether to discard a cluster based on inter-cluster distance.

**Other results.** Due to space constraints, we have included additional results in the appendix. Specifically, Appendix A.5 evaluates the impact of data distribution on the performance of existing defenses, Appendix A.6 assesses MARS's sensitivity to hyperparameters, and Appendix A.7 examines MARS's effectiveness on larger datasets such as ImageNet Deng et al. (2009).

## 6 CONCLUSION

We propose MARS, a malignity-aware backdoor defense. Unlike existing defenses that rely on loosely backdoor-coupled empirical statistical metrics, MARS directly focuses on the core nature of backdoor attacks by detecting malignity through the backdoor energies of neurons. We further amplify this malignity by extracting the most prominent backdoor energies. A novel Wasserstein-based clustering method is then introduced to accurately detect backdoored models. Comprehensive comparisons across 3 datasets, 3 SOTA attacks, and 8 SOTA defenses demonstrate the superiority of MARS. Moreover, we validate the robustness of MARS against adaptive attack, further showcasing its effectiveness in backdoor defense.

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

# A APPENDIX

## A.1 PROOF OF THEOREM 1

**Theorem 1 (Upper Bound of Backdoor Energy).** *Suppose a L-layer neural network $F$ and its every sub-network $f^{(l)}, l \in [1, L]$, are Lipschitz smooth. Then, the backdoor energy of the $k^{th}$ neuron in the $l^{th}$ layer can be upper bounded by:*

$$BE_k^{(l)}(F) \leq \|f_k^{(l)}\|_{\text{Lip}} \left( \prod_{i=1}^{l-1} \|f^{(i)}\|_{\text{Lip}} \right) \mathbb{E}_{x \sim \mathcal{X}} \left[ \|x - \delta(x)\|_2 \right],$$

*where $\|.\|_{Lip}$ represents the Lipschitz constant of a function.*

*Proof:*

Since each sub-network $f^{(i)}, i \in [1, L]$ is Lipschitz smooth, for all $x, y$, we have:

$$\|f^{(i)}(x) - f^{(i)}(y)\|_2 \leq \|f^{(i)}\|_{\text{Lip}} \|x - y\|_2$$

Consider the difference in the $k$-th neuron of the $l$-th layer between clean and backdoor inputs:

$$\begin{aligned}
\|F_k^{(l)}(x) - F_k^{(l)}(\delta(x))\|_2 &= \|f_k^{(l)} \circ f^{(l-1)} \circ \cdots \circ f^{(1)}(x) - f_k^{(l)} \circ f^{(l-1)} \circ \cdots \circ f^{(1)}(\delta(x))\|_2 \\
&\leq \|f_k^{(l)}\|_{\text{Lip}} \|f^{(l-1)} \circ \cdots \circ f^{(1)}(x) - f^{(l-1)} \circ \cdots \circ f^{(1)}(\delta(x))\|_2 \\
&\leq \|f_k^{(l)}\|_{\text{Lip}} \|f^{(l-1)}\|_{\text{Lip}} \|f^{(l-2)} \circ \cdots \circ f^{(1)}(x) - f^{(l-2)} \circ \cdots \circ f^{(1)}(\delta(x))\|_2 \\
&\leq \cdots \\
&\leq \|f_k^{(l)}\|_{\text{Lip}} \left( \prod_{i=1}^{l-1} \|f^{(i)}\|_{\text{Lip}} \right) \|x - \delta(x)\|_2
\end{aligned}$$

In the proof above, we apply the Lipschitz smooth assumption layer by layer from the outermost to the innermost layers of the network. When considering an outer layer, all remaining inner sub-networks are treated as a single entity.

Taking the expectation over $x \sim \mathcal{X}$:

$$BE_k^{(l)}(F) = \mathbb{E}_{x \sim \mathcal{X}} \left[ \|F_k^{(l)}(x) - F_k^{(l)}(\delta(x))\|_2 \right] \leq \|f_k^{(l)}\|_{\text{Lip}} \left( \prod_{i=1}^{l-1} \|f^{(i)}\|_{\text{Lip}} \right) \mathbb{E}_{x \sim \mathcal{X}} \left[ \|x - \delta(x)\|_2 \right]$$

Thus, we conclude the proof.

$\square$

## A.2 ALGORITHM DESCRIPTION

Algorithm 1 provides a detailed description of MARS. The central server first calculates the backdoor energy (BE) for all neurons in each local model (Lines 1-6), then extracts the most prominent BE values from each layer to form the concentrated backdoor energy (CBE), which is stored in a set $A$ (Line 7). Using the Wasserstein-based clustering algorithm, the server clusters all local models' CBEs into two groups based on the CBEs in set $A$, storing the client indices of each cluster in $S_1$ and $S_2$, respectively (Line 8). The centers $A_1$ and $A_2$ of the two clusters are computed (Lines 9-10). If the Wasserstein distance between $A_1$ and $A_2$ is within an acceptable threshold $\epsilon$, it indicates that the distributions of the two clusters are similar, and thus all local models are considered benign (Lines 11-12). Otherwise, the local models corresponding to the cluster with the smaller norm of the cluster center are used for aggregation (Lines 13-18).

## A.3 EVALUATED ATTACKS AND DEFENSES

### A.3.1 ATTACKS

**MRA Bagdasaryan et al. (2020).** MRA (Model Replacement Attack) is the first backdoor attack specifically designed for FL. Its core idea is to amplify backdoor updates in proportion, allowing a

---

**Algorithm 1:** MARS

---

**Input** : Set of selected client in the current round: $S$;
       Set of corresponding local models: $\{F(.;\theta_s), s \in S\}$;     ▷ We omit round index $t$ for simplicity
       Top factor: $\kappa$;
       Inter-cluster threshold: $\epsilon$.
**Output:** Aggregated global model: $F(.;\theta^G)$.

1  $A \leftarrow \{\}$                     ▷ Set $A$ is used to preserve the CBE of local models
2  **for** $s \in S$ **do**
3     $\theta \leftarrow \theta_s$
4     **for** $l \in L$ **do**
5         **for** $k \in n_l$ **do**
6            $BE_k^{(l)}(F(.;\theta)) = \|f_k^{(l)}\|_{Lip}$         ▷ Calculate BE of each neuron via eq. 3
7     $A[s] \leftarrow \bigcup_{l=1}^{L} TopK_{\kappa\%}\left(\{BE_i^{(l)}(F(.;\theta))\}_{i=1}^{n_l}\right)$   ▷ Calculate CBE for each client via eq. 4
8  $S_1, S_2 \leftarrow$ **K-WMeans**(A)              ▷ Divide client index into two clusters $S_1$ and $S_2$
9  $A_1 \leftarrow$ **Mean**$(\{A[s], s \in S_1\})$
10  $A_2 \leftarrow$ **Mean**$(\{A[s], s \in S_2\})$
11  **if** $Wass(A_1, A_2) < \epsilon$ **then**
12     $S_{final} \leftarrow S$                  ▷ All the global models are used for aggregation
13  **else**
14     **if** $\|A_1\|_1 < \|A_2\|_1$ **then**
15         $S_{final} \leftarrow S_1$            ▷ Preserve the cluster with lower norm of central CBE
16     **else**
17         $S_{final} \leftarrow S_2$
18  $\theta^G \leftarrow \frac{1}{|S_{final}|}\sum_{s \in S_{final}} \theta_s$         ▷ Aggregate all the credible local model weights
19  **return** $F(.;\theta^G)$

---

small number of malicious updates to dominate the global model. MRA is widely used for assessing the robustness of backdoor defenses in FL.

**CerP Lyu et al. (2023).** CerP (Cerberus Poisoning) is an advanced backdoor attack algorithm for FL that has emerged in recent years. It simultaneously tunes the backdoor trigger while controlling the changes in the poisoned model for each malicious participant, enabling a stealthy yet effective backdoor attack against a wide range of FL defense mechanisms. By fine-tuning the trigger, CerP increases the compatibility between the backdoor model and the trigger, minimizing significant updates to the model parameters. Additionally, controlling the changes in the model reduces the disparity between the backdoor and benign models, making it more challenging for defenders to identify the backdoor models.

**3DFed Li et al. (2023).** 3DFed is an adaptive and extensible framework designed for launching covert backdoor attacks in FL environments, particularly in a black-box setting. It addresses the challenges posed by existing backdoor attacks, which often require extensive information about the victim FL system and typically optimize for a single objective, rendering them less effective against sophisticated defense mechanisms. The core of 3DFed lies in its three evasion modules that effectively camouflage backdoor models: backdoor training with constrained loss, noise mask, and decoy model. These components work synergistically to implant indicators into the backdoor model, allowing 3DFed to capture attack feedback from the global model during the previous training epoch. This feedback enables dynamic adjustment of hyper-parameters within the evasion modules, enhancing the stealth and efficacy of the attacks. To the best of our knowledge, MARS is the first defense to conduct a comprehensive evaluation of robustness against 3DFed.

### A.3.2 DEFENSES

**FedAvg McMahan et al. (2017).** FedAvg is the first aggregation algorithm for FL, which constructs a high-performance global model by aggregating all local models through weighted averaging. Due to its effective knowledge aggregation capabilities, FedAvg is widely utilized in real-world industrial

applications, such as Google's GBoard. Consequently, existing works usually evaluate the resistance of FedAvg to backdoor attacks, making it a critical baseline for comparison.

**Multi-Krum Blanchard et al. (2017).** Multi-Krum is a defense algorithm based on out-of-distribution (OOD) detection. It estimates whether a local model deviates from the overall distribution by calculating the sum of distances between that model and its nearest $n - f - 2$ neighbor models ($n$ and $f$ represent the number of participants and the number of attackers, respectively). Subsequently, it excludes the models that are furthest from the overall distribution from the aggregation queue.

**RFLBAT Wang et al. (2022).** RFLBAT is a cutting-edge defense mechanism designed to counteract backdoor attacks in FL systems. Unlike existing algorithms that often impose constraints on the number of malicious attackers or assume independent and identically distributed (IID) data, RFLBAT operates effectively under realistic conditions where the number of attackers is unknown and the data distribution is typically non-IID. RFLBAT leverages principal component analysis (PCA) to identify and extract essential features from the model updates, followed by a K-means clustering algorithm to group similar updates. This dual approach enables RFLBAT to effectively filter out malicious updates without requiring additional auxiliary information beyond the learning process itself.

**FLAME Nguyen et al. (2022).** FLAME is a defense framework aimed at countering backdoor attacks in FL. The key implementation steps of FLAME are as follows: *Noise Estimation.* FLAME estimates the optimal amount of noise to inject, ensuring effective elimination of backdoors while preserving model performance. *Model Clustering.* The framework utilizes a clustering approach to group similar models, which helps identify and isolate potentially malicious updates. *Weight Clipping.* FLAME applies weight clipping to the clustered models, mitigating the influence of adversarial updates and maintaining the integrity of the aggregated model. Through these steps, FLAME effectively defends against backdoor attacks with minimal impact on the performance of benign updates.

**FoolsGold Fung et al. (2020).** FoolsGold is a consistency detection-based defense that identifies poisoning updates based on the diversity of client updates in the distributed learning process. Specifically, Updates with excessively high pairwise cosine similarity are assigned lower aggregation weights. Unlike prior work, FoolsGold does not bound the expected number of attackers, requires no auxiliary information outside of the learning process, and makes fewer assumptions about clients and their data.

**FLDetector Zhang et al. (2022).** FLDetector is a defense mechanism designed to address the challenge of model poisoning attacks in FL, particularly when there is a large number of malicious clients. The core insight behind FLDetector is that model poisoning attacks lead to inconsistent updates from malicious clients across multiple iterations. To identify these inconsistencies, FLDetector predicts each client's model update in subsequent iterations based on its historical updates and flags a client as malicious if its updates deviate from the predicted values across several iterations. This approach allows FLDetector to accurately detect and remove malicious clients, ensuring that existing robust FL methods can continue to function effectively even under strong attack scenarios.

**DeepSight Rieger et al. (2022).** DeepSight is a model filtering approach designed to mitigate backdoor attacks in FL without removing benign models from clients with diverse data distributions. Unlike existing defenses that simply exclude deviating models, DeepSight introduces three novel techniques to better characterize the data distribution behind model updates and measure subtle differences in the internal structure and outputs of neural networks (NNs). These techniques allow DeepSight to detect suspicious model updates effectively. Additionally, it employs a clustering scheme to group models and identify clusters that contain poisoned updates with high attack impact. By combining these insights, DeepSight can eliminate harmful model clusters, while also mitigating any residual backdoor effects using weight clipping defenses.

**FedCLP Zheng et al. (2022).** CLP (Lipschitzness based Pruning) is a novel approach designed to detect and remove backdoor channels in deep neural networks (DNNs) without requiring any data. It introduces the concept of the Channel Lipschitz Constant (CLC), which measures the Lipschitz constant of the mapping from input images to the output of each channel. By analyzing the correlation between an upper bound of the CLC (UCLC) and the activation changes caused by a backdoor trigger, CLP identifies potential backdoor channels. Since UCLC can be directly computed from

the network's weight matrices, CLP operates in a completely data-free manner. Once these infected channels are detected, CLP prunes them to repair the model. This method is fast, simple, and robust, making it an efficient solution for backdoor defense with minimal dependency on the choice of the pruning threshold. We adapt CLP to the FL setting and name it FedCLP. Specifically, we prune each local model using CLP to remove backdoor-related information before aggregating them with FedAvg.

**BackdoorIndicator Li & Dai (2024).** BackdoorIndicator is a proactive backdoor detection mechanism specifically designed for FL systems. This mechanism operates on the insight that deploying subsequent backdoors with the same target label can enhance the accuracy of existing backdoors. BackdoorIndicator enables the server to inject indicator tasks into the global model using out-of-distribution (OOD) data. Since any backdoor samples are inherently OOD concerning benign samples, the server, unaware of the specific backdoor types or target labels, can effectively detect backdoor presence in uploaded models by evaluating the performance of these indicator tasks. Through comprehensive empirical evaluations, BackdoorIndicator demonstrates consistently superior performance and practicality compared to existing baseline defenses across various system configurations and adversarial scenarios.

## A.4 EVALUATION METRICS

We evaluate the performance of a defense using four metrics: ACC, ASR, TPR, and FPR, each providing distinct perspectives on the effectiveness of the defense. Additionally, based on these metrics, we introduce a novel metric called CAD, which offers a comprehensive view of the overall effectiveness of the defense.

**ACC.** ACC (Model Accuracy) is calculated as the proportion of correctly identified clean samples to the total number of clean samples. In federated learning, maintaining high accuracy is crucial, as it reflects the model's overall performance in making correct predictions across all clients.

**ASR.** ASR (Attack Success Rate) measures the proportion of samples with triggers that are classified as the target label. A lower ASR indicates that the defense mechanism is effective in identifying and mitigating backdoor attacks. In federated learning scenarios, minimizing ASR is essential to ensure the system remains resilient against adversarial manipulation. It is important to note that in our experiments, we do not exclude samples corresponding to the target label. As a result, even for a clean model, the ASR does not approach 0 but rather tends toward $1/c$ ($c$ represents the total number of classes).

**TPR.** TPR (True Positive Rate) measures the proportion of backdoor models that are correctly identified by the defense algorithm as malicious. High TPR is indicative of the defense algorithm's effectiveness in accurately detecting backdoor models. A robust defense mechanism should achieve high TPR to minimize the risk of allowing backdoor attacks to compromise the integrity of the model. This is critical for maintaining trust and reliability in federated learning environments.

**FPR.** FPR (False Positive Rate) measures the proportion of legitimate models that are incorrectly classified by the defense algorithm as backdoored. A low FPR is crucial as it indicates that the defense algorithm does not mistakenly flag benign models as backdoored. In the context of federated learning, minimizing FPR is essential to prevent unnecessary disruptions to legitimate model updates and to maintain the overall functionality of the system.

**CAD.** CAD (Comprohensive Abilisty of Defense) is a composite metric that integrates the four aforementioned indicators to provide an overall assessment of a defense algorithm's performance. It is calculated as follows:

$$CAD = \frac{ACC + (1 - ASR) + TPR + (1 - FPR)}{4} \times 100\%. \tag{7}$$

This formulation captures a balanced view of accuracy, attack resistance, true positive detection, and false positive minimization.

It is important to note that FoolsGold does not directly discard local models but assigns lower aggregation weights to suspected models. When calculating its TPR and FPR, we consider local models with an aggregation weight greater than $0.5$ as selected by FoolsGold, otherwise, the model is deemed rejected. Additionally, FedCLP does not distinguish between benign and backdoor models,

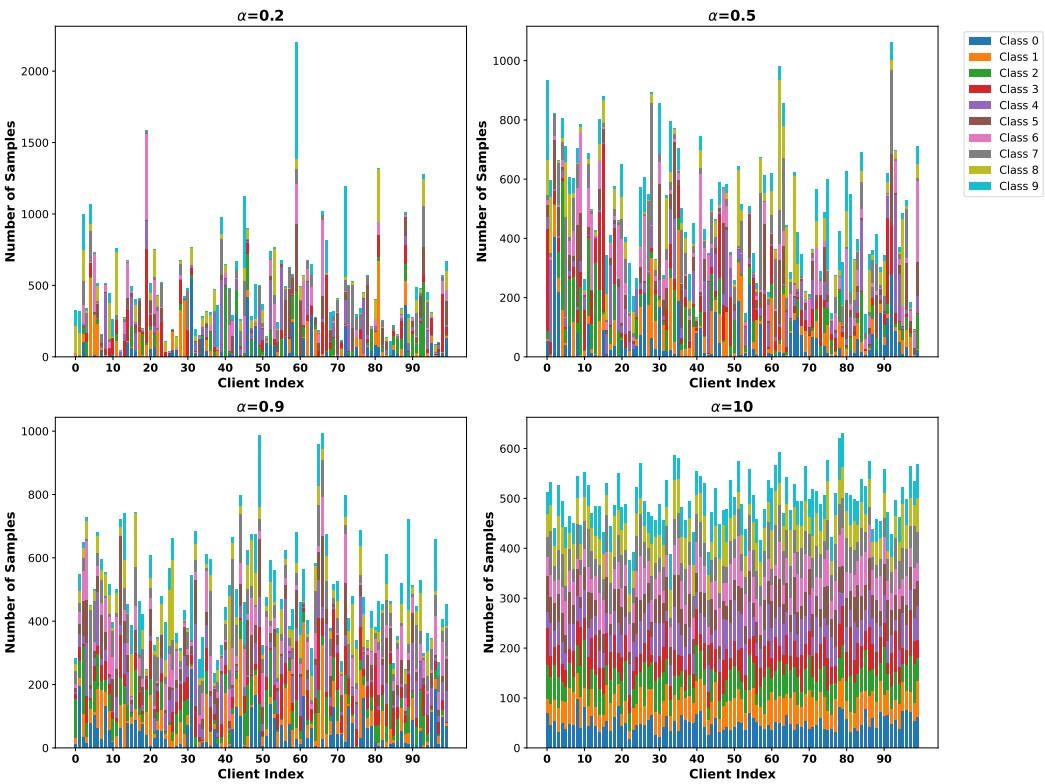

Figure 3: Dirichlet sampling with different $\alpha$.

instead pruning all local models before aggregation. Therefore, TPR and FPR cannot be calculated for FedCLP, and we denote the values of these metrics as "-". For the CAD calculation of FedCLP, we only consider ACC and ASR, *i.e.*, $CAD = \frac{ACC+(1-ASR)}{2} \times 100\%$.

## A.5   IMPACT OF DATA DISTRIBUTION

The previous experiments are conducted using a Dirichlet sampling parameter of $\alpha = 0.9$, which is the default setting recommended by 3DFed. To assess the impact of a broader range of data distributions on the performance of existing defenses, we follow BackdoorIndicator by considering three non-IID data distributions with $\alpha$ values of 0.2, 0.5, and 0.9. Notably, a smaller $\alpha$ indicates a higher degree of data heterogeneity. Additionally, we examine an IID data distribution ($\alpha = 10$), a scenario often overlooked by existing defenses. Figure 3 illustrates the data distribution of each client under different values of alpha. As shown in Table 6, overall, the performance of existing defenses gradually deteriorates as data heterogeneity increases. For instance, FLAME effectively counters 3DFed attacks with a CAD of $86.09\%$ at $\alpha = 10$, but it completely fails in non-IID scenarios, with a CAD of only around $30\%$. While FedCLP can mitigate backdoor attacks, it also leads to varying degrees of ACC reduction, with more significant drops in non-IID settings. Interestingly, we observe a counterintuitive phenomenon where FLDetector performs worse in IID scenarios; we speculate that this is because 3DFed makes fewer modifications to the backdoor models in IID settings, making the predicted models and backdoor models more similar, which causes FLDetector to mistakenly classify benign models as backdoor models. MARS consistently performs excellently across all data distributions, with a CAD always above $93\%$.

## A.6   SENSITIVITY TO HYPERPARAMETERS

### A.6.1   IMPACT OF DISTANCE METRIC

In Section 4.5, we illustrate with a toy example that Wasserstein distance is more suitable for MARS compared to traditional Euclidean and cosine distances. To further substantiate our claim, we replace MARS's distance metric with Euclidean distance and cosine distance, keeping all other components

Table 6: Impact of data distribution on the performance of exitsting defenses under 3DFed attack on CIFAR-10.

| $\alpha$ | Metric | FedAvg | MultiKrum | RFLBAT | FLAME | FoolsGold | FLDetector | DeepSight | FedCLP | MARS |
|---|---|---|---|---|---|---|---|---|---|---|
| | ACC | 83.38 | 82.47 | 82.82 | 80.08 | 82.66 | 82.04 | 76.65 | 60.87 | 83.26 |
| | ASR | 97.37 | 99.44 | 95.75 | 97.22 | 93.97 | 93.48 | 98.32 | 4.72 | 9.38 |
| 0.2 | TPR | 0.00 | 17.50 | 0.00 | 0.00 | 30.00 | 0.00 | 0.00 | - | 100.00 |
| | FPR | 0.00 | 20.63 | 0.00 | 56.25 | 60.63 | 0.00 | 43.75 | - | 0.00 |
| | CAD | 46.50 | 44.98 | 46.77 | 31.65 | 39.51 | 47.14 | 33.65 | 78.07 | 93.47 |
| | ACC | 84.24 | 83.88 | 83.58 | 83.41 | 83.60 | 84.89 | 83.89 | 73.28 | 84.66 |
| | ASR | 98.39 | 96.40 | 97.49 | 96.72 | 98.28 | 91.79 | 90.11 | 10.28 | 9.90 |
| 0.5 | TPR | 0.00 | 0.00 | 0.00 | 0.00 | 0.00 | 0.00 | 0.00 | - | 100.00 |
| | FPR | 0.00 | 25.00 | 0.00 | 56.25 | 68.75 | 0.00 | 6.25 | - | 0.00 |
| | CAD | 46.46 | 40.62 | 46.52 | 32.61 | 29.14 | 48.28 | 46.88 | 81.50 | 93.69 |
| | ACC | 84.37 | 84.07 | 84.30 | 83.06 | 84.11 | 84.24 | 84.80 | 69.25 | 85.07 |
| | ASR | 96.76 | 97.27 | 92.02 | 97.50 | 96.29 | 95.20 | 98.85 | 7.55 | 9.86 |
| 0.9 | TPR | 0.00 | 0.00 | 0.00 | 2.50 | 0.00 | 0.00 | 0.00 | - | 100.00 |
| | FPR | 0.00 | 25.00 | 5.00 | 55.63 | 0.25 | 35.00 | 6.25 | - | 0.00 |
| | CAD | 46.90 | 40.45 | 46.82 | 33.11 | 46.89 | 38.51 | 44.93 | 80.85 | 93.80 |
| | ACC | 84.60 | 84.51 | 85.19 | 85.28 | 84.64 | 83.82 | 84.67 | 76.20 | 85.30 |
| | ASR | 95.68 | 98.54 | 76.18 | 9.67 | 96.36 | 99.13 | 70.08 | 8.75 | 9.49 |
| 10 | TPR | 0.00 | 5.00 | 20.00 | 100.00 | 0.00 | 0.00 | 0.00 | - | 100.00 |
| | FPR | 0.00 | 23.75 | 11.25 | 31.25 | 0.00 | 100.00 | 6.25 | - | 0.00 |
| | CAD | 47.23 | 41.81 | 54.44 | 86.09 | 47.07 | 21.17 | 52.09 | 83.73 | 93.95 |

constant. As shown in Table 7, both Euclidean and cosine distances fail to accurately detect backdoor updates, resulting in a CAD of only around $44\%$. In contrast, when using Wasserstein distance, MARS achieves optimal performance with a CAD close to $94\%$. This supports our hypothesis that Wasserstein distance, which is insensitive to the order of elements, is more effective for detecting backdoor models in our scenario.

Table 7: Impact of distance metric on MARS under CerP attack on CIFAR-10.

| Dist. | ACC $\uparrow$ | ASR $\downarrow$ | TPR $\uparrow$ | FPR $\downarrow$ | CAD $\uparrow$ |
|---|---|---|---|---|---|
| Euc. | 83.93 | 88.15 | 35.00 | 51.25 | 44.88 |
| Cos. | 84.29 | 82.05 | 32.50 | 58.13 | 44.15 |
| Wass. | 85.37 | 10.03 | 100.00 | 0.00 | 93.84 |

### A.6.2 SENSITIVITY TO $\epsilon$

In Section 4.5, to avoid blindly removing a cluster in non-adversarial scenarios, which could degrade model accuracy, we propose using inter-cluster distance to decide whether to retain all clusters, with an acceptable threshold set to $\epsilon$. As shown in Table 8, in the presence of attackers, MARS accurately distinguishes between benign and malicious models as long as $\epsilon$ does not exceed 1. In non-adversarial scenarios, when $\epsilon$ is no less than 0.03, MARS does not mistakenly classify any benign models as backdoor models. Therefore, setting $\epsilon$ between 0.03 and 1 ensures optimal performance for MARS. The wide range of acceptable $\epsilon$ values indicates that MARS is not highly sensitive to this parameter, making it easy to select an appropriate $\epsilon$ in real-world scenarios.

Table 8: Impact of $\epsilon$ on MARS under 3DFed attack on CIFAR-10.

| | Metric | 0.01 | 0.02 | 0.03 | 0.04 | 0.05 | 0.10 | 0.50 | 1.00 | 3.00 | 5.00 |
|---|---|---|---|---|---|---|---|---|---|---|---|
| w/ attack | TPR | 100.00 | 100.00 | 100.00 | 100.00 | 100.00 | 100.00 | 100.00 | 100.00 | 63.64 | 18.18 |
| | FPR | 0.00 | 0.00 | 0.00 | 0.00 | 0.00 | 0.00 | 0.00 | 0.00 | 0.00 | 0.00 |
| w/o attack | FPR | 42.73 | 8.18 | 0.00 | 0.00 | 0.00 | 0.00 | 0.00 | 0.00 | 0.00 | 0.00 |

### A.6.3 SENSITIVITY TO $\kappa$

Review that in Section 4.4, in order to concentrate backdoor activity and facilitate subsequent detection of backdoor models, we extract the top $\kappa\%$ of BE values from each layer of local models,

forming a one-dimensional vector called CBE. As shown in Table 9, when $\kappa$ is set to 10 or less, MARS achieves a TPR of $100\%$ and an FPR of $0\%$, indicating that MARS can precisely detect all backdoor models without mistakenly discarding any benign models. However, when $\kappa$ exceeds 20, MARS begins to miss some backdoor models, and in some cases, even misidentifies a few benign models as backdoor ones. In real-world deployments, setting $\kappa$ to 10 or below ensures optimal performance (with the default in this paper being 5), which is easily achievable. Therefore, MARS is not highly sensitive to the choice of $\kappa$.

Table 9: Impact of $\kappa$ on MARS under 3DFed attack on CIFAR-10.

| Metric | 1 | 2 | 5 | 10 | 20 | 40 | 60 | 80 | 100 |
|--------|-------|-------|-------|-------|-------|-------|-------|-------|-------|
| TPR | 100.00 | 100.00 | 100.00 | 100.00 | 94.44 | 87.10 | 83.33 | 91.67 | 77.42 |
| FPR | 0.00 | 0.00 | 0.00 | 0.00 | 0.35 | 0.00 | 0.42 | 0.52 | 0.00 |

## A.7 PERFORMANCE ON IMAGENET

In the main text, we evaluate the effectiveness of MARS on MNIST, CIFAR-10, and CIFAR-100, following the common practice in existing defenses such as BackdoorIndicator and FLDetector. However, real-world datasets are typically more complex and challenging. Hence, it is essential to assess the performance of MARS on larger, more intricate datasets. We use ImageNet as the benchmark dataset and ReXNet as the network architecture. Regarding attacks, due to the lack of open-source code compatible with ImageNet for 3DFed and CerP, and after several attempts to adapt their parameters to work with ImageNet without success, we focus solely on the MRA attack. On the defense side, we compare MARS with FedAvg in both adversarial and non-adversarial (referred to as the Baseline) settings. As shown in Table 10, with FedAvg, ASR escalates from $0.14\%$ to $98.54\%$ as training progresses, highlighting the significant threat posed by MRA to federated learning systems. However, when MARS is deployed on the central server, ACC remains consistently above $75\%$, and ASR is reduced to around $0.1\%$, comparable to the Baseline. This demonstrates that MARS is effective even when applied to large-scale datasets like ImageNet.

Table 10: Comparison of MARS under MRA attack on ImageNet.

| Round | Defense | ACC ↑ | ASR ↓ | TPR ↑ | FPR ↓ | CAD ↑ |
|-------|---------|-------|-------|-------|-------|-------|
| 1 | FedAvg | 69.54 | 0.14 | 0.00 | 0.00 | 67.35 |
| | MARS | 75.87 | 0.10 | 100.00 | 0.00 | 93.94 |
| | Baseline | 76.25 | 0.08 | - | - | - |
| 10 | FedAvg | 74.64 | 1.05 | 0.00 | 0.00 | 68.40 |
| | MARS | 75.47 | 0.12 | 100.00 | 0.00 | 93.84 |
| | Baseline | 75.85 | 0.08 | - | - | - |
| 20 | FedAvg | 73.81 | 19.94 | 0.00 | 0.00 | 63.47 |
| | MARS | 75.44 | 0.12 | 100.00 | 0.00 | 93.83 |
| | Baseline | 75.89 | 0.08 | - | - | - |
| 30 | FedAvg | 73.91 | 84.12 | 0.00 | 0.00 | 47.45 |
| | MARS | 75.49 | 0.12 | 100.00 | 0.00 | 93.84 |
| | Baseline | 75.59 | 0.08 | - | - | - |
| 40 | FedAvg | 74.19 | 95.59 | 0.00 | 0.00 | 44.65 |
| | MARS | 75.22 | 0.12 | 100.00 | 0.00 | 93.78 |
| | Baseline | 75.34 | 0.08 | - | - | - |
| 50 | FedAvg | 73.73 | 98.54 | 0.00 | 0.00 | 43.80 |
| | MARS | 75.14 | 0.12 | 100.00 | 0.00 | 93.76 |
| | Baseline | 75.26 | 0.08 | - | - | - |

