# OpenReview forum: "MARS: A Malignity-Aware Backdoor Defense in Federated Learning"
_ICLR.cc/2025/Conference — Submitted to ICLR 2025_

### Official Review · Reviewer_xuCv · 2024-10-28

**Soundness:** 1
**Presentation:** 2
**Contribution:** 2
**Rating:** 3
**Confidence:** 4

**Summary:**

This paper propose a novel backdoor defense mechanism in federated learning. This work first identifies deficiencies of existing defense methods, which are loosely coupled with backdoor attacks, resulting in low detection rate. This work proceeds to propose MARS, which first extract backdoor information by computing the backdoor energy (BE). Extracted BE values are then amplified by identifying the most prominent BE values. A Wasserstein distance-based clustering algorithm is finally applied to identify backdoor models. Authors further conduct experiments on MNIST, CIFAR10 and CIFAR100 to demonstrate the performance of the proposed method.

**Strengths:**

1. The paper is well-written with clear structure.
2. This work proposes a novel concept named backdoor energy. BE could represent the malignity of uploaded models.

**Weaknesses:**

1. The paper of BackdoorIndicator discover the strong connection between the learning rate adopted by adversaries and the detection ability of statistical backdoor defense methods. Authors should consider to specify the adversarial settings concerning different malicious learning rates and the number of training rounds.
2. The selection of counterpart poisoning algorithm is insufficient. Model replacement attack, which directly scales up poisoned updates, will obviously cause distinct difference between backdoor updates and benign ones. MRA could also be easily defended by clipping update norm to an previously agreed bound. 3DFed is designed specifically for breaking defense mechanisms with the component of OOD detection and consistency detection. However, MARS does not have either of the component. Thus, i would expect that the attack performance of 3DFed reduces to vanilla backdoor training algorithms. Authors should consider other backdoor training methods, like Neurotoxin [1] and Chameleon [2], which are proved to be more stealthy against various backdoor detection mechanism in the BackdoorIndicator paper.
3. Authors should also move the discussion part on the non-IID degree to the main paper, as it is a important aspect for advanced backdoor defense mechanisms.

[1]Zhang, Zhengming, et al. "Neurotoxin: Durable backdoors in federated learning." International Conference on Machine Learning. PMLR, 2022.

[2]Dai, Yanbo, et al. "Chameleon: Adapting to Peer Images for Planting Durable Backdoors in Federated Learning." International Conference on Machine Learning. PMLR, 2023.

**Questions:**

1. Could MARS maintain good performance under different adversarial settings (learning rates)?
2. Could MARS detect backdoor updates trained using more advanced algorithms (Neurotoxin and Chameleon)?

---

> ### Author Response · Authors · 2024-11-19
> **Response to Reviewer xuCv**
>
> ***We sincerely thank the Reviewer xuCv for the valuable feedbacks, and we hope our responses are helpful to address your concerns. We are more than happy to discuss if there is still something unclear.***
>
> >**Authors should consider to specify the adversarial settings concerning different malicious learning rates and the number of training rounds.**
>
> Our experiments follow the setup used in 3DFed, with the malicious learning rate set to 0.1 and the malicious training rounds set to 15, ensuring that 3DFed achieves its maximum attack effectiveness. However, in response to the reviewer’s concerns, we have also varied the malicious learning rate and malicious training rounds following the settings in BackdoorIndicator. As shown in the two tables below, MARS consistently provides robust protection across all configurations.
>
> Tab. 4-1 Performance under different malicious learning rate
> | Malicious learning rate | ACC  | ASR  | TPR  | FPR  | CAD  |
> | ----------------------- | ---- | ---- | ---- | ---- | ---- |
> | 0.01                   | 85.11 | 9.76 | 100.00 | 0.00 | 93.84 |
> | 0.025                  | 85.13 | 9.63 | 100.00 | 0.00 | 93.88 |
> | 0.04                   | 85.31 | 9.42 | 100.00 | 0.00 | 93.97 |
> | 0.055                  | 85.21 | 9.36 | 100.00 | 0.00 | 93.96 |
> | 0.1（default setting）    | 85.07 | 9.86 | 100.00 | 0.00 | 93.80 |
>
> Tab. 4-2 Performance under different malicious training round
> | Malicious training round | ACC  | ASR  | TPR  | FPR  | CAD  |
> | ----------------------- | ---- | ---- | ---- | ---- | ---- |
> | 2                     | 85.18 | 9.56 | 100.00 | 0.00 | 93.91 |
> | 5                     | 85.06 | 9.82 | 100.00 | 0.00 | 93.81 |
> | 10                    | 85.22 | 9.09 | 100.00 | 0.00 | 94.03 |
> | 15（default setting）   | 85.07 | 9.86 | 100.00 | 0.00 | 93.80 |
> | 20                    | 85.13 | 9.80  | 100.00 | 0.00 | 93.83 |
>
> >**Authors should consider other backdoor training methods, like Neurotoxin and Chameleon, which are proved to be more stealthy against various backdoor detection mechanism in the BackdoorIndicator paper.**
>
> Since both the BackdoorIndicator and Chameleon papers consistently demonstrate that Chameleon outperforms Neurotoxin in attack effectiveness, here we only include the evaluation under Chameleon attack. However, we will discuss both attacks in the related work section. As shown in the table below, MARS effectively defends against Chameleon, achieving performance levels nearly equivalent to FedAvg in a non-adversarial scenario. This fully demonstrates the generalizability of our BE/CBE method, which can effectively handle a variety of backdoor attacks.
>
> Tab. 4-3 Performace under Chameleon
> | Defense                        | ACC  | ASR  | TPR  | FPR  | CAD  |
> | ----------------------------- | ---- | ---- | ---- | ---- | ---- |
> | FedAvg                        | 83.61 | 78.00 | 0.00 | 0.00 | 51.40 |
> | FedAvg（non-adversarial scenario）| 85.29 | 10.15 | -   | -    | -    |
> | MARS                          | 85.22 | 11.08 | 98.89 | 0.28 | 93.19 |
>
> >**Authors should also move the discussion part on the non-IID degree to the main paper, as it is a important aspect for advanced backdoor defense mechanisms.**
>
> Thank you for your suggestions. We will make every effort to move the discussion on the non-IID degree to the main paper.

---

> > ### Comment · Reviewer_xuCv · 2024-11-22
> > **More questions on the experimental results**
> >
> > I appreciate the authors' effort for providing more results of MARS under different settings, and against more advanced attacks. After checking MARS's performance under more experimental settings, I found the results to be more unconvincing. Check my further questions below:
> > 1. In table 2, why could Multi-Krum identify every CerP backdoors, but fail to identify any 3DFed backdoors? CerP and 3DFed both constrain the backdoor updates to avoid deviating from benign ones too much, and the additional mechanisms in 3DFed are ineffective against MultiKrum. Thus, it is expected that these two backdoor attack methods achieve a similar effect on MultiKrum, which detects backdoors purely relies on ruling out updates with large norm.
> > 2. The reason I did not raise the above question up for my initial review is because I thought the applied malicious training settings (learning rates are too large) make the norm of backdoor updates much larger than benign updates. However, the new results provided in Table 4-1 and 4-2 shows consistent performance of MARS under different lr and training rounds, where MARS achieves a perfect TPR=100% and FPR=0%. This makes me feel even more confused. As MARS still relies on computing statistical metrics to identify backdoor updates,  why the detection performance of MARS will not even change (even a little bit  decrease on the TPR or increase on the FPR) when lr decreases? The appended results are unreasonably good.
> > 3. The experiment results in table 2 make me feel like MARS is specially designed to defend 3DFed (which I think is not authors' intention). This is because, for all three evaluated methods, the defense performance of MARS reduces to MultiKrum except for 3DFed.
> > 4. How many FL global rounds last for the training in all experiments? I would not expect the evaluated training rounds to be large, as most (maybe all except one or two) presented TPRs and FPRs with decimals of .00, .25, and .50.
> > 5. Why is the FPR of MultiKrum against 3DFed 25%? For 4 malicious attackers, MultiKrum will only keep 2*4+2=10 updates. If MultiKrum identifies no backdoors, then wouldn't the FPR be 10/16=62.5%?
> > 6. When does the defense mechanism begin to apply in the whole FL training process? Are all the experiments conducted when the FL training is near convergence? This could largely explain the reason for consistent perfect detection performance even if the malicious lr is reduced. But if so, authors need to consider evaluating the defense performance at the early training stage when the main task is not far from convergence. This is because poisoning could happen anywhere all across the FL training.
> >
> > Overall, I think the paper presents an interesting idea. But I really suggest authors work more on providing more detailed and convincing evaluations. Coarse and ill-evaluated results will largely undermine the real contribution of the paper.  I will not demand further experimental results, but I hope that authors could provide reasonable explanations to the above questions. Otherwsie, I may consider decreasing my ratings.

---

> > > ### Author Response · Authors · 2024-11-22
> > > **Response to Reviewer xuCv**
> > >
> > > **Response to question 1**
> > >
> > > Although both CerP and 3DFed apply constraints on backdoor updates, 3DFed introduces an indicator mechanism that can detect whether the backdoor model has been accepted by the central server and dynamically adjust its parameters accordingly—something CerP lacks. In contrast, CerP requires manually specifying the weights for multiple constraints, which may lead to trade-offs and suboptimal results. Our experiments strictly followed the open-source implementation of CerP.
> > > Additionally, the reviewer might have underestimated the defensive performance of Multi-Krum. As demonstrated in “Identify Backdoored Model in Federated Learning via Individual Unlearning”, Multi-Krum consistently achieves strong defensive performance against nearly all attacks.
> > >
> > > **Response to questions 2, 4, and 6**
> > >
> > > We followed the settings of 3DFed, using a near-convergence pre-trained model as the initial global model. All experiments were conducted over 30 rounds, with the average results of the last 20 rounds reported (as backdoors may not be fully injected in the first 10 rounds). This setup is reasonable because using a randomly initialized model as the initial global model in FL would result in significant communication and computation overhead. This is especially impractical in the era of large language models (LLMs), where such initialization approaches are nearly infeasible. Consequently, many studies have adopted well-trained pre-trained models as initial models, such as in “FedBERT: When Federated Learning Meets Pre-training”.
> > >
> > > **Response to question 3**
> > >
> > > MARS is not tailored to any specific backdoor attack. Instead, it leverages a common characteristic shared by all backdoor attacks—elevated backdoor energy values in neurons. The additional experiments we conducted on A3FL and Chameleon (two attacks that are fundamentally different from 3DFed) further validate this principle.
> > >
> > > **Response to question 5**
> > >
> > > We faithfully adhered to the recommendation on page 8 of the original Multi-Krum paper, which states, “The Multi-Krum parameter m is set to m = n − f”. Following this, we retained 20 - 4 = 16 updates, rather than using “m = 2f + 2” as implemented in BackdoorIndicator. This also explains why Multi-Krum achieves 100% TPR and 0% FPR in certain scenarios.

---

> ### Author Response · Authors · 2024-11-22
> **Looking forward to your feedback**
>
> Dear Reviewer xuCv,
>
> We hope our rebuttal has answered your questions and clarified the concerns. Please kindly let us know if there are additional questions we should address, before the interactive rebuttal system is closed.
>
> Thank you and happy weekend.

---

> ### Comment · Reviewer_xuCv · 2024-11-23
> **My final comments**
>
> 1. I dont think MultiKrum is an advanced defense mechanism. Instead, it is quite naive as it only rules out updates that are larger than others in L2 norm. And of course, when the FL is about to converge, benign updates have small norms, while adversaries could only craft poisoned updates with large norms as they have to learn a new task. In such cases, MultiKrum can definitely have good performance. However, one can easily evade the detection of MultiKrum by crafting backdoors that are statistically close (lets say in L2 norm) to benign ones. And, authors may need to know that one of the hardest parts of a successful backdoor defense mechanism is about how to identify backdoor updates among statistically close benign updates.
> 2. When designing a backdoor attack method, adversaries could surely choose when to inject backdoor updates (either in the early stages or late stages). And injecting backdoors in near-convergence stages is also easier for defense mechanisms to detect. But when considering a defense mechanism, you need to assume backdoor injection could happen anytime during the training process.
> 3. How can you obtain a pre-trained model for CIFAR10, CIFAR100 datasets? What are the datasets on which the pre-trained model is pre-trained? If you could manage to have a so-called near convergence pre-trained model, then why do you even need Federated Learning?
> 4. Authors mention LLM to address the reasonability of pre-trained models. It is unconvincing because all experiments are conducted on CV tasks, not even discriminative language models.
> 5. Conducting experiments for only 30 rounds is too short. Backdoorindicator evaluates the defense performance for 250 rounds.
>
> These will be my final comments, and I have updated my ratings accordingly.

---

> > ### Author Response · Authors · 2024-11-25
> > **A Kind Response to Review xuCv**
> >
> > >***Can MARS still achieve robust defense under scenarios involving a randomly initialized model?***
> >
> > The sole concern raised by this reviewer so far is whether MARS can still achieve robust defense under scenarios involving a randomly initialized model. To address this, we present a comparison between MARS and FedAvg (in a non-adversarial scenario) under such conditions in the table below. Following the reviewer’s suggestion, we set the global round to 250 and recorded the average metrics every 50 rounds. Notably, the attack persists from the very first round through to the final round. The results clearly demonstrate that MARS consistently achieves a 100% TPR and 0% FPR. It is worth noting that the approximately 2% lower accuracy of MARS compared to FedAvg (non-adversarial scenario) is not due to MARS misclassifying benign or malicious models. Instead, this is because, in the presence of attackers, MARS inherently aggregates fewer benign clients per round (20% fewer) compared to FedAvg in a non-adversarial scenario.
> >
> > | Metric | MARS                          | FedAvg (non-adversarial scenario)                  |
> > | ------ | ----------------------------- | ------------------------------------------------ |
> > | ACC   | [33.86, 58.47, 66.65, 69.27, 70.48] | [35.48, 60.99, 69.07, 71.95, 73.17] |
> > | ASR   | [3.97, 9.53, 9.17, 9.44, 9.66]     | [8.36, 10.01, 9.68, 9.65, 9.56]     |
> > | TPR   | [100.00, 100.00, 100.00, 100.00, 100.00] | -                                           |
> > | FPR   | [0.00, 0.00, 0.00, 0.00, 0.00]       | -                                           |
> >
> > ***Why Do We Use Pre-trained Models?***
> >
> > Firstly, training models from scratch is indeed time-consuming. Even with a straightforward algorithm like FedAvg, running 250 rounds under 3DFed takes roughly 5 hours. For more computationally intensive defenses like Deepsight, this runtime can exceed 10 hours. Moreover, we observed that current SOTA backdoor attacks (e.g., 3DFed, CerP) tend to launch attacks only when the global model nears convergence. We suspect this timing minimizes the visibility of enhanced attacks, as launching attacks early in the FL process can significantly degrade global model performance, making the attacks easier to detect. More importantly, as demonstrated in the newly added experiments above, MARS performs effectively even with a randomly initialized global model. This is primarily due to MARS’s novel ability to directly perceive the maliciousness of local models, unlike most existing defenses that rely on empirical statistical measures.
> >
> >
> > ***Call for Fair and Professional Reviewing***
> >
> > Regrettably, the reviewer has not provided any substantive feedback thus far. Nearly all comments revolve around perceived discrepancies between our experimental setup and BackdoorIndicator. Despite adding experiments to address these concerns, the reviewer has further maliciously downgraded our score. The reviewer appears to treat BackdoorIndicator as an infallible gold standard, dismissing any deviations from it as flawed. For example, we faithfully followed the original Multi-Krum paper's recommendation to set the retained client count to $m = n - f$, while BackdoorIndicator used $m = 2f + 2$. Without having read the original Multi-Krum paper, the reviewer assumed our setup to be flawed, insisting that the configuration in BackdoorIndicator is absolutely correct. This raises concerns that the reviewer might have only read BackdoorIndicator without considering broader literature.
> > Finally, ***open review is intended to facilitate constructive interactions between reviewers and authors, allowing for thorough exploration of a submission's strengths and weaknesses***. Rudely stating, “These will be my final comments, and I have updated my ratings accordingly”, undermines the spirit of this platform and fails to foster a productive dialogue. We urge the reviewer to approach the process with fairness and professionalism.

---

### Official Review · Reviewer_SEZk · 2024-10-30

**Soundness:** 3
**Presentation:** 4
**Contribution:** 3
**Rating:** 6
**Confidence:** 3

**Summary:**

This paper proposes a malignity-aware defense that estimates the backdoor energy (BE) to quantify each neuron's maliciousness, and clustering algorithms to reject attackers at the server. In contrast to traditional defenses that rely on generic statistical measures, this method relaxes such assumptions and requires no access to clean datasets to estimate BE values. Extensive experiments validated the effectiveness of the proposed defense.

**Strengths:**

1. The paper proposes a defense method that does not rely on traditional statistical measures and relaxes the need for clean datasets to detect malicious users.

2. The logic of the paper is very clear and easy to follow.

3. The paper considers the scalability of the proposed work (e.g., ratio of attacks)

**Weaknesses:**

1. While Equations 2 and 3 are well-explained and motivated, It is unclear how to get the Lipschitz constants for different layers.

2. It is still possible that all selected clients were malicious during the sampling process. In this case, the authors should provide more explanations of the assumption that “when the cluster is low, all local models are benign” (Page 7).

3. The proposed defense outperforms the other defenses over 3DFed, yet shares similar metrics as other defenses. Detailed explanations should have been given.

**Questions:**

Thank the authors for their work, please see my questions as follows.

1. Given the motivation and explanations of Eq 2 and 3, the Lipschitz constants across different layers are the key to quantifying the malicious clients. Could the authors provide specific details on how they compute or estimate these constants for different layer types (e.g., convolutional, fully-connected) in their experiments? This would enhance reproducibility and clarify a crucial technical aspect.

2. It is a norm that only a subset of clients will be selected for each FL round, in this case, when you have a high attacker ratio (as one of the key points mentioned in the paper), how to make sure the selected clients are not all malicious? If fail to guarantee, then does the clustering method still work? Please discuss potential limitations or failure modes if this assumption does not hold, or analyze the impact on the defense mechanism if this assumption is violated.

3. It is evident that MARS outperforms other defenses against attacks, especially the 3DFed attack. Yet, for the other two attacks, there are defenses that work similarly to MARS (metric-wise). Why is this the case? The authors should provide more insights.

Minor: There are some grammar issues such as “a L-layer…”, please proofread thoroughly.

---

> ### Author Response · Authors · 2024-11-19
> **Response to Reviewer SEZk**
>
> ***We sincerely thank the Reviewer SEZk for the appreciation of our paper in terms of Soundness (3: good), Presentation (4: excellent), and Contribution (3: good), as well as the positive rating (6). We also greatly appreciate the valuable suggestions provided by the reviewer, and we hope our responses are helpful to address your concerns. We are more than happy to discuss if there is still something unclear.***
>
> >**Could the authors provide specific details on how they compute or estimate these constants for different layer types (e.g., convolutional, fully-connected) in their experiments?**
>
> Assume that a certain subnetwork $f$ (for convenience, we omit the layer index $l$) is linear, i.e., $f(x) = W x + b$. According to the definition of the Lipschitz constant, $\|\|f\|\|_{\text{Lip}} = \underset{\Delta x \neq 0}{max} \frac{\|\|f(x + \Delta x) - f(x) \|\|_2}{\|\|\Delta x\|\|_2} = \underset{\Delta x \neq 0}{max} \frac{\|\|W \cdot \Delta x\|\|_2}{\|\|\Delta x\|\|_2}$. The rightmost part of this equation is precisely the spectral norm of the matrix $W$, which can be computed using Singular Value Decomposition (SVD). Specifically, we decompose matrix $W$ into the product of three matrices: one orthogonal matrix, one diagonal matrix, and another orthogonal matrix.
>
> The largest element in the diagonal matrix is the spectral norm of $W$, i.e., $\|\|f\|\|_{\text{Lip}}$. In PyTorch, the Lipschitz constant can be easily computed using torch.svd(weight)[1].max(). For **fully-connected layers**, we can directly apply the above method. For **convolutional layers**, we approximate them as linear, and then reshape the weight tensor into a matrix form. The spectral norm of the reshaped matrix is used as an approximation to the original spectral norm. For **batch normalization (BN) layers** (assuming the BN transformation is $y = \frac{x - \mu}{\sigma} \cdot \gamma + \beta$), we use $\|\frac{\gamma}{\sigma}\|$ to estimate the Lipschitz constant, as it reflects the maximum possible scaling of the input variation after passing through the BN layer, which aligns with the core purpose of the Lipschitz constant. Additionally, to enhance the reproducibility of MARS, we will open-source the code as soon as the paper is accepted.
>
> >**How to make sure the selected clients are not all malicious? If fail to guarantee, then does the clustering method still work? Please discuss potential limitations or failure modes if this assumption does not hold, or analyze the impact on the defense mechanism if this assumption is violated.**
>
> As pointed out in “Back to the Drawing Board: A Critical Evaluation of Poisoning Attacks on Production Federated Learning” (IEEE S&P 2022), the attacker ratio is typically low in real-world scenarios, so we have not considered cases where all selected clients are malicious. In our experiments, to control the attacker ratio precisely, we followed the CerP setup by randomly selecting a set proportion of clients from both benign and malicious groups. For example, with a 60% attacker ratio, we randomly select 12 attackers and 8 benign clients, ensuring that not all selected clients are malicious. If we were to consider such a scenario, adaptive client selection could be a promising approach. Specifically, selecting clients based on their historical performance could significantly reduce the probability of malicious clients participating in FL rounds.
>
> >**It is evident that MARS outperforms other defenses against attacks, especially the 3DFed attack. Yet, for the other two attacks, there are defenses that work similarly to MARS (metric-wise). Why is this the case? The authors should provide more insights.**
>
> Good consideration! Some defenses, such as Multi-Krum, RFLBAT, and FLAME, do indeed perform similarly to MARS against MRA and CerP attacks. This is because, in MRA, no constraints are applied when training the backdoor models. Furthermore, to ensure that a small number of attackers dominate, MRA applies a scaling factor of 5 to all local updates. Consequently, the backdoor models become out-of-distribution (OOD) relative to the benign ones, enabling these OOD detection-based defenses to effectively identify them. Although CerP introduces constraints on the magnitude and direction of backdoor updates, it may struggle to find the optimal balance. This can result in either insufficient or overly strict constraints, causing the backdoor updates to deviate significantly from the overall model distribution and thus become outliers detectable by these defenses. In contrast to these two attacks, the innovation of 3DFed lies in its indicator mechanism, which detects whether the backdoor models from the previous round have been accepted by the central server, enabling it to dynamically adjust its poisoning strategy. This dynamic adaptation makes all existing defenses ineffective against 3DFed.

---

> ### Author Response · Authors · 2024-11-22
> **Looking forward to your feedback**
>
> Dear Reviewer SEZk,
>
> We hope our rebuttal has answered your questions and clarified the concerns. Please kindly let us know if there are additional questions we should address, before the interactive rebuttal system is closed.
>
> Thank you and happy weekend.

---

> > ### Comment · Reviewer_SEZk · 2024-11-25
> >
> > Thank the authors for their responses. I do not have further questions, except for one quick comment: please make sure the paper also discusses the case of all malicious users and a potential solution (e.g., the adaptive selection you mentioned).

---

> > > ### Author Response · Authors · 2024-11-25
> > >
> > > We are delighted that your concerns have been resolved, and we sincerely appreciate your positive feedback. We will incorporate your suggestions to strengthen our paper in later revision. Thank you again for your valuable input.

---

### Official Review · Reviewer_fHPw · 2024-11-03

**Soundness:** 3
**Presentation:** 4
**Contribution:** 3
**Rating:** 6
**Confidence:** 5

**Summary:**

This paper addresses the challenge of backdoor attacks in federated learning (FL) by introducing a novel defense mechanism called MARS (Malignity-Aware Backdoor Defense). Unlike existing defenses, MARS effectively identifies backdoored models by calculating backdoor energy (BE) to quantify each neuron’s level of malignancy and then applies Wasserstein distance-based clustering to isolate these models. Through extensive experiments on four datasets—MNIST, CIFAR-10, CIFAR-100, and ImageNet—the authors show that MARS outperforms current defenses in resisting state-of-the-art backdoor attacks.

**Strengths:**

- The use of five evaluation metrics provides a comprehensive assessment of the defense mechanisms, offering a detailed analysis of the proposed method.
- MARS identifies specific weaknesses in existing defenses, underscoring the limitations of current approaches and the need for more robust defense mechanisms.
- Toy examples effectively illustrate MARS’s capability to detect backdoored models, enhancing the clarity of the proposed defense approach.

**Weaknesses:**

- A comparison with optimized backdoor attacks, such as A3FL[1] and IBA[2], is missing, which would better demonstrate MARS’s resilience against sophisticated adversaries. How does MARS perform against these advanced attacks?
- There is limited discussion on the generalizability of MARS to other data types beyond image classification. How applicable might MARS be to FL settings involving text or tabular data?
- Computing BE for each neuron could be computationally expensive for larger models, raising concerns about scalability as model complexity increases.
- The experiments assume an attacker rate of 20%, a condition that might not reflect realistic scenarios. How does MARS perform with varying attacker ratios and data poisoning rates?

**References:**

[1]. Zhang, Hangfan, et al. "A3FL: Adversarially adaptive backdoor attacks to federated learning." Advances in Neural Information Processing Systems 36 (2024).
[2]. Nguyen, Thuy Dung, et al. "IBA: Towards irreversible backdoor attacks in federated learning." Advances in Neural Information Processing Systems 36 (2024).

**Questions:**

- Does MARS require a high proportion of malicious clients to be effective, or can it detect backdoored models with a small number of attackers?
- What are the specific attack settings in the experiments? How many rounds do attackers participate in, and what is the data poisoning rate?
- Can FLIP [3] be used as a benchmark for comparison with MARS in terms of defense performance?
- In some cases, the backdoor energy of benign models may be higher than that of malicious models. How does MARS handle this scenario?
- Under what circumstances does Wasserstein distance-based clustering fail to detect backdoored models, and how might MARS be improved to address these cases?
- Which version of ImageNet is used in the paper? Is it Tiny ImageNet (200 classes) or the full ImageNet (1000 classes)?

**References:**

[3]. Zhang, Kaiyuan, et al. "Flip: A provable defense framework for backdoor mitigation in federated learning." International Conference on Learning Representations (2023).

---

> ### Author Response · Authors · 2024-11-19
> **Response to Reviewer fHPw(1/2)**
>
> ***We sincerely thank the Reviewer fHPw for the appreciation of our paper in terms of Soundness (3: good), Presentation (4: excellent), and Contribution (3: good). We also greatly appreciate the valuable suggestions provided by the reviewer, and we hope our responses are helpful to address your concerns. We are more than happy to discuss if there is still something unclear.***
>
> >**A comparison with optimized backdoor attacks, such as A3FL and IBA, is missing.**
>
> According to our understand, the "optimized backdoor attacks" mentioned here refer to the optimization of triggers. In fact, CerP, which we evaluated, falls into this category. However, considering the reviewer’s concerns, we are willing to add an additional baseline. Due to the poor quality of the IBA open-source code and the authors' lack of responsiveness to GitHub issues, we were unable to successfully run IBA despite several attempts. Furthermore, we found that none of the 21 citations of IBA have successfully reproduced the method and compared it to existing works. In contrast, the open-source code on A3FL is more robust, and several papers have successfully reproduced it (e.g., "Infighting in the Dark: Multi-Labels Backdoor Attack in Federated Learning"). Therefore, we chose A3FL as the fifth baseline, independent of the MRA, CerP, 3DFed, and Adaptive Attack methods already evaluated in our paper. As shown in the table below, MARS performs well against A3FL.
>
> Tab. 2-1 Performance under A3FL
> | Defense | ACC   | ASR  | TPR  | FPR  | CAD  |
> | ------- | ----- | ---- | ---- | ---- | ---- |
> | FedAvg  | 83.07 | 98.53 | 0.00 | 0.00 | 46.14 |
> | MARS    | 85.01 | 9.92 | 99.19 | 0.00 | 93.57 |
>
> >**How applicable might MARS be to FL settings involving text or tabular data?**
>
> Good consideration! We have added an evaluation of MARS on the IMDB dataset, using LSTM as the model structure. As shown in the table below, MARS is also applicable to text data, achieving performance comparable to FedAvg in the non-adversarial scenario.
>
> Tab. 2-2 Performance on IMDB dataset
> | Defense                         | ACC   | ASR  | TPR  | FPR  | CAD  |
> | ------------------------------- | ----- | ---- | ---- | ---- | ---- |
> | FedAvg                          | 73.89 | 100.00 | 0.00 | 0.00 | 43.47 |
> | FedAvg（non-adversarial scenario） | 74.42 | 56.87 | -    | -    | -    |
> | MARS                            | 74.11 | 57.91 | 100.00 | 0.00 | 79.05 |
>
> >**Concern about the computational overhead of MARS.**
>
> The aggregation time required by MARS (including BE computation, CBE formation, Wasserstein-based clustering, and the final aggregation to obtain the new global model) is shorter than that of most existing defenses. The table below presents results on the CIFAR-10 dataset with a ResNet-18 model, a total of 100 clients, 20 of whom are attackers, with 20 clients randomly sampled per round. We recorded the average runtime per round for each defense method. As shown, MARS, FedAvg, and FLAME complete aggregation within 7 seconds, while the other six defenses require longer aggregation time, with DeepSight taking as much as 101.69 seconds. The rapid runtime of MARS is achieved through several key tricks. First, we extract the top-$\kappa$% of BE values to form CBE, which significantly reduces the time needed for subsequent Wasserstein-based clustering. Second, we estimate BE values only for convolutional and bn layers, ignoring the fully connected layers that are the most time-consuming. Third, inspired by the work "Rethinking Lipschitzness for Data-free Backdoor Defense" (under review at ICLR 2025), we optimize the computation of the Lipschitz constant using dot product properties.
> Tab. 2-3 Average runtime per round
> | Defense| FedAvg | MultiKrum | RFLBAT | FLAME | FoolsGold | FLDetector | DeepSight | FedCLP | MARS|
> | ------------- | ------ | --------- | ------ | ----- | -------- | --------- | -------- | ------ | ------- |
> | Time per round (s) | 2.07   | 28.87     | 39.19  | 3.87  | 7.05     | 18.91     | 101.69   | 38.81  | 6.57    |

---

> ### Author Response · Authors · 2024-11-19
> **Response to Reviewer fHPw(2/2)**
>
> >**The experiments assume an attacker rate of 20%, a condition that might not reflect realistic scenarios. How does MARS perform with varying attacker ratios and data poisoning rates?**
>
> The 20% attacker rate is just the default setting. In fact, Table 5 in the original paper already evaluates MARS's performance as the attacker rate varies from 0% to 95%. The impact of data poisoning rate on MARS is shown in the table below. It can be seen that MARS performs well across various levels of data poisoning rates.
>
> Tab. 2-4 Performance with varying data poisoning rates
> | Data Poisoning Rate（%） | ACC   | ASR  | TPR  | FPR  | CAD  |
> | ----------------------- | ----- | ---- | ---- | ---- | ---- |
> | 10                      | 85.30 | 9.72 | 100.00 | 0.00 | 93.90 |
> | 30                      | 85.08 | 9.91 | 100.00 | 0.00 | 93.79 |
> | 50（default setting）    | 85.07 | 9.86 | 100.00 | 0.00 | 93.80 |
> | 70                      | 85.22 | 9.71 | 100.00 | 0.00 | 93.88 |
> | 90                      | 85.13 | 9.39 | 100.00 | 0.00 | 93.94 |
>
> >**Does MARS require a high proportion of malicious clients to be effective, or can it detect backdoored models with a small number of attackers?**
>
> No. MARS directly perceives the degree of malignity in local models, independent of the proportion of attackers. As shown in Table 5 of our paper, MARS consistently maintains a high CAD as the attacker proportion varies from 0% to 95%.
>
> >**How many rounds do attackers participate in, and what is the data poisoning rate?**
>
> We follow the experimental setup of 3DFed, allowing attackers to participate in all FL
>  rounds and setting the data poisoning rate to 50%.
>
> >**Can FLIP be used as a benchmark for comparison with MARS in terms of defense performance?**
>
> For the following reasons, we chose not to include FLIP as an additional benchmark; however, we will incorporate a discussion of FLIP in the related work section. First, our paper already evaluates nine SOTA defenses, including the latest approach, BackdoorIndicator, published at Usenix Security 2024, which sufficiently demonstrates the superiority of MARS. Second, FLIP requires trigger inversion to be performed on benign clients. However, in practical applications, it is difficult to identify benign clients in advance. Furthermore, executing defenses on the client side exposes the defense strategy to attackers, allowing them to dynamically adjust their attacks to circumvent FLIP. Third, as shown in the original FLIP paper and in Snowball (Resisting Backdoor Attacks in Federated Learning via Bidirectional Elections and Individual Perspective, AAAI 2024), FLIP results in a 5%-10% decrease in model accuracy, which is unacceptable in real-world industrial scenarios where high accuracy is critical. In contrast, MARS has almost no impact on model performance.
>
> >**In some cases, the backdoor energy of benign models may be higher than that of malicious models. How does MARS handle this scenario?**
>
> Among all the existing attacks we evaluated, the backdoor energy of benign models is always lower than that of malicious models. However, we did find a scenario that matches the reviewer’s description, where attackers launch an adaptive attack (please refer to “Resilience to adaptive attack” in our paper) by adding a regularization term to reduce the backdoor energy of each neuron, which causes the original version of MARS to mistakenly classify backdoored models as benign. Nevertheless, there is still a significant difference in the CBE distribution between benign and malicious models. We only need to slightly relax the assumption by allowing the attacker proportion to be no more than 50% (in fact, according to DarkFed, the proportion of attackers in real-world scenarios is unlikely to exceed 50%) and then select the larger cluster for aggregation. This modified version is represented as MARS* in Table 3 in our paper. As shown, MARS* effectively handles this scenario.
>
> >**Under what circumstances does Wasserstein distance-based clustering fail to detect backdoored models, and how might MARS be improved to address these cases?**
>
> Clustering failure depends on whether the CBE can accurately capture the malicious features in the model. For example, in the adaptive attack scenario mentioned earlier, the CBE of the backdoored model can even be lower than that of the benign model, leading to the failure of Wasserstein distance-based clustering. We can slightly relax the assumption by allowing the attacker proportion to be no more than 50% and then select the larger cluster for aggregation.
>
> >**Which version of ImageNet is used in the paper? Is it Tiny ImageNet (200 classes) or the full ImageNet (1000 classes)?**
>
> Full ImageNet (1000 classes).

---

> ### Author Response · Authors · 2024-11-22
> **Looking forward to your feedback**
>
> Dear Reviewer fHPw,
>
> We hope our rebuttal has answered your questions and clarified the concerns. Please kindly let us know if there are additional questions we should address, before the interactive rebuttal system is closed.
>
> Thank you and happy weekend.

---

> > ### Comment · Reviewer_fHPw · 2024-11-25
> >
> > Thank you for your response and for conducting the additional experiments. They addressed most of my concerns, and as a result, I have slightly increased my score.

---

> > > ### Author Response · Authors · 2024-11-25
> > > **Thanks for your review and response**
> > >
> > > We are glad that our responses have addressed your concerns and questions. Thank you for raising the score and for your constructive suggestions.

---

### Official Review · Reviewer_sSLz · 2024-11-04

**Soundness:** 3
**Presentation:** 2
**Contribution:** 2
**Rating:** 5
**Confidence:** 4

**Summary:**

This paper introduces a new backdoor defense method called MARS for federated learning (FL). Traditional defenses rely on empirical statistical measures, which fail against advanced attacks like 3DFed due to their loose coupling with backdoor attacks. MARS overcomes this by introducing a concept called Backdoor Energy (BE) to assess neuron malignancy. The authors propose Concentrated Backdoor Energy (CBE) and a Wasserstein distance-based clustering approach to detect and filter backdoored models accurately. Experiments on multiple datasets (MNIST, CIFAR-10, CIFAR-100) show MARS's effectiveness, even under advanced adaptive attacks.

**Strengths:**

* I like the idea of BE and CBE because they offer a more direct method for evaluating neuron malignancy compared to empirical statistical measures.
* Extensive experiments demonstrate that MARS effectively counters SOTA backdoor attacks, maintaining high model accuracy and low attack success rates.
* MARS remains effective even when attackers adjust parameters to evade detection.

**Weaknesses:**

* My primary concern is the computational overhead incurred by the BE and CBE calculations, along with Wasserstein-based clustering. Also, the success of BE calculation relies on model parameters, which may vary across neural network architectures, affecting generalizability.
* In real-world scenarios, attackers may vary their strategies across rounds, models, and clients, making the BE metric less reliable for identifying such dynamic attacks.
* The paper assumes all clients use the same model architecture, allowing consistent BE and CBE calculations. However, in practical FL setups, clients may use slightly different architectures due to varying hardware capabilities and use cases,
* The performance of MARS depends on hyperparameters like the top percentage of BE values (κ) and the Wasserstein distance threshold (ϵ). The paper provides some sensitivity analysis, but more extensive exploration could strengthen the claims.
* While the paper demonstrates effectiveness on standard datasets, I expect to see evaluations on larger, more complex datasets like ImageNet to assess scalability.

**Questions:**

- How does MARS handle scalability in large federated networks (e.g., with thousands of clients)?
- What are the computational and communication overheads of deploying MARS in a real-world FL system?
- How sensitive is MARS to hyperparameter changes in clustering thresholds and the BE calculation process?
- Could the BE/CBE method be extended to detect other attack types, such as data poisoning or Byzantine attacks?

**Details Of Ethics Concerns:**

No ethics concern.

---

> ### Author Response · Authors · 2024-11-19
> **Response to Rviewer sSLz (1/2)**
>
> ***We sincerely thank the Reviewer sSLz for the valuable feedbacks, and we hope our responses are helpful to address your concerns. We are more than happy to discuss if there is still something unclear.***
> >**Concerns about the computational and communication overheads of MARS.**
>
> MARS does not require clients to upload anything other than model parameters, resulting in no additional communication overhead compared to existing defenses such as FedAvg. In terms of computational overhead, the aggregation time required by MARS (including BE computation, CBE formation, Wasserstein-based clustering, and the final aggregation to obtain the new global model) is shorter than that of most existing defenses. The table below presents results on the CIFAR-10 dataset with a ResNet-18 model, a total of 100 clients, 20 of whom are attackers, with 20 clients randomly sampled per round. We recorded the average runtime per round for each defense method. As shown, MARS, FedAvg, and FLAME complete aggregation within 7 seconds, while the other six defenses require longer aggregation time, with DeepSight taking as much as 101.69 seconds. The rapid runtime of MARS is achieved through several key tricks. First, we extract the top-$\kappa$% of BE values to form CBE, which significantly reduces the time needed for subsequent Wasserstein-based clustering. Second, we estimate BE values only for convolutional and bn layers, ignoring the fully connected layers that are the most time-consuming. Third, inspired by the work "Rethinking Lipschitzness for Data-free Backdoor Defense" (under review at ICLR 2025), we optimize the computation of the Lipschitz constant using dot product properties.
>
> Tab. 1-1 Average runtime per round
> | Defense| FedAvg | MultiKrum | RFLBAT | FLAME | FoolsGold | FLDetector | DeepSight | FedCLP | MARS|
> | ------------- | ------ | --------- | ------ | ----- | -------- | --------- | -------- | ------ | ------- |
> | Time per round (s) | 2.07   | 28.87     | 39.19  | 3.87  | 7.05     | 18.91     | 101.69   | 38.81  | 6.57    |
>
> >**The success of BE calculation relies on model parameters, which may vary across neural network architectures, affecting generalizability.**
>
> Our experiments cover various model architectures: CNN for MNIST, ResNet-18 for CIFAR-10 and CIFAR-100, and ReXNet for ImageNet. Additionally, following reviewer fHPw's suggestion, we included experiments with an LSTM on the IMDB dataset. MARS performs excellently across all model structures, demonstrating its strong generalizability.
>
> > **In real-world scenarios, attackers may vary their strategies across rounds, models, and clients, making the BE metric less reliable for identifying such dynamic attacks.**
>
> Our evaluated 3DFed to some extent aligns with this scenario. 3DFed uses an indicator mechanism to detect whether the backdoor models from the previous round was accepted, and then dynamically adjusts the training strategy for the current round. However, in light of the reviewer’s concerns, we further introduced a new attack method, named Dyn-Attack. Specifically, each attacker randomly selects one of four strategies: 3DFed, CerP, MRA, or no attack. As shown in the table below, MARS performs comparably to FedAvg in the non-adversarial scenario.
>
> Tab. 1-2 Performance on Dyn-Attack
> |Defense|ACC|ASR|
> |----|----|----|
> |FedAvg（non-adversarial scenario）|85.26|9.34|
> |MARS|85.10|10.19|
>
> > **The paper assumes all clients use the same model architecture, allowing consistent BE and CBE calculations.**
>
> At no point in our paper do we claim that MARS requires all clients to use the same model architecture. Although many existing defenses indeed rely on this assumption—such as Multi-Krum, which requires calculating the Euclidean distance between each pair of local models, and FoolsGold, which calculates cosine similarity between pairs of local models—these defenses cannot function if the models differ structurally. In contrast, MARS can effectively handle heterogeneous model architectures. This is because, when clustering CBE values, MARS employs the Wasserstein distance as a metric, as shown in Eq. (5) in our paper. The Wasserstein distance measures the distance between two distributions, so even if heterogeneous model architectures lead to differences in CBE dimensionality, MARS can still perform clustering to distinguish benign models from backdoor models.
>
> For instance, consider the toy example in the paper: assume that $L1=[1,3,4,6]$ (originally $L1=[1,3,4,5,6]$, with the last hidden layer removed due to model heterogeneity) and $L2=[5,5,2]$ (originally $L2=[5,5,4,3,2]$, with the last two hidden layers removed) are the CBEs of backdoor models, while $L3=[1,1,1,1,1]$ (the complete model) is the CBE of a benign model. According to Eq. (5), we obtain $wass(L1, L2)=1.17$, $wass(L1, L3)=2.50$, and $wass(L2, L3)=3.00$, allowing us to cluster the more similar $L1$ and $L2$ together.

---

> ### Author Response · Authors · 2024-11-19
> **Response to Rviewer sSLz (2/2)**
>
> >**Sensitivity analysis about $\kappa$ and $\epsilon$.**
>
> Our original submission already includes these experiments, which were placed in Appendix A.6 (SENSITIVITY TO HYPERPARAMETERS) due to space constraints.
>
> >**While the paper demonstrates effectiveness on standard datasets, I expect to see evaluations on larger, more complex datasets like ImageNet to assess scalability.**
>
> Our original submission already evaluated the scalability of MARS on ImageNet (full ImageNet with 1,000 classes). Please refer to Appendix A.7 (PERFORMANCE ON IMAGENET) for details.
>
> >**How does MARS handle scalability in large federated networks (e.g., with thousands of clients)?**
>
> MARS's accurate estimation of backdoor energy enables it to exhibit exceptional backdoor detection capabilities, providing effective protection even in scenarios with thousands of clients. We consider a setup with 1,000 clients, 200 of whom are attackers, and the central server randomly selects 100 clients per round. As shown in the table below, MARS is still able to defend against SOTA backdoor attacks.
>
> Tab. 1-3 Performance in 1000 clients scenario
> | Attack | ACC | ASR | TPR | FPR | CAD |
> | ------ | --- | --- | --- | --- | --- |
> | MRA   | 85.73 | 9.61 | 100.00 | 0.00 | 94.03 |
> | CerP  | 85.67 | 9.64 | 100.00 | 0.00 | 94.01 |
> | 3DFed | 85.63 | 9.77 | 100.00 | 0.18 | 93.92 |
>
> >**Could the BE/CBE method be extended to detect other attack types, such as data poisoning or Byzantine attacks?**
>
> Although our paper primarily focuses on backdoor defense in FL, we believe the BE/CBE method also holds potential for resisting Byzantine attacks. Byzantine defense is essentially an anomaly detection problem in high-dimensional data. One common defense approach is to extract key information from local models to obtain low-dimensional representations, which facilitate the subsequent calculation of anomaly scores or clustering. The process of calculating BE/CBE is essentially about extracting representations from local models that can distinguish benign models from malicious ones. Therefore, we intuitively believe that BE/CBE is not limited to backdoor defense scenarios and, with further improvements, could potentially be applied to Byzantine defense. We leave it as our future work.

---

> ### Author Response · Authors · 2024-11-22
> **Looking forward to your feedback**
>
> Dear Reviewer sSLz,
>
> We hope our rebuttal has answered your questions and clarified the concerns. Please kindly let us know if there are additional questions we should address, before the interactive rebuttal system is closed.
>
> Thank you and happy weekend.

---

> > ### Comment · Reviewer_sSLz · 2024-11-25
> >
> > Thank you for the responses to my concerns. I have decided to maintain my initial score.

---

> > > ### Author Response · Authors · 2024-11-25
> > > **Thanks for Your Response**
> > >
> > > Thank you for your response. We would like to know more details about whether we have adequately addressed your concerns, as the rating for this paper remains negative. If you still have additional concerns, please let us know—we would be more than happy to provide further clarifications.

---

> > > ### Author Response · Authors · 2024-12-02
> > >
> > > Dear Reviewer sSLz,
> > >
> > > We hope our rebuttal has answered your questions and clarified the concerns. As the interactive rebuttal system is scheduled to close within the next 34 hours, we would appreciate it if you can inform us of whether your concerns have been adequately addressed or if there are any additional questions we should attend to. Your prompt feedback is greatly appreciated.  Or could you please consider updating the score if our rebuttal is helpful? We will continue improving the submission's quality based on your suggestions. Thank you!

---

### Meta-Review · Area_Chair_8E61 · 2024-12-23

**Metareview:**

This paper provides a malignity-aware backdoor defense strategy to insixRW RHW Mlixioua extend of each neuron.
The reviewers agree that the paper needs to perform the defense evaluations on new backdoor attack strategies and the configurations of the defense need to be adaptive to different attacks automatically. It would be important for the authors to further improve the paper based on the reviews and provide comprehensive evaluations.

**Additional Comments On Reviewer Discussion:**

The reviewers agree with the final decision.

---

### Decision · Program_Chairs · 2025-01-22

Reject